# Imprinted *Dlk1* dosage as a size determinant of the mammalian pituitary gland

Valeria Scagliotti[1], Maria Lillina Vignola[1], Thea Willis[1,2], Mark Howard[3], Eugenia Marinelli[1], Carles Gaston-Massuet[4], Cynthia Andoniadou[2], Marika Charalambous[1]*

[1]Department of Medical and Molecular Genetics, Faculty of Life Sciences and Medicine, King's College London, London, United Kingdom; [2]Centre for Craniofacial and Regenerative Biology, Faculty of Dentistry, Oral and Craniofacial Sciences, King's College London, London, United Kingdom; [3]MRC Centre for Transplantation, Peter Gorer Department of Immunobiology, School of Immunology & Microbial Sciences, King's College London, London, United Kingdom; [4]Centre for Endocrinology, William Harvey Research Institute, Barts and the London School of Medicine and Dentistry, Queen Mary University of London, London, United Kingdom

**Abstract** Co-regulated genes of the Imprinted Gene Network are involved in the control of growth and body size, and imprinted gene dysfunction underlies human paediatric disorders involving the endocrine system. Imprinted genes are highly expressed in the pituitary gland, among them, *Dlk1*, a paternally expressed gene whose membrane-bound and secreted protein products can regulate proliferation and differentiation of multiple stem cell populations. Dosage of circulating DLK1 has been previously implicated in the control of growth through unknown molecular mechanisms. Here we generate a series of mouse genetic models to modify levels of *Dlk1* expression in the pituitary gland and demonstrate that the dosage of DLK1 modulates the process of stem cell commitment with lifelong impact on pituitary gland size. We establish that stem cells are a critical source of DLK1, where embryonic disruption alters proliferation in the anterior pituitary, leading to long-lasting consequences on growth hormone secretion later in life.

**\*For correspondence:**
marika.charalambous@kcl.ac.uk

**Competing interest:** The authors declare that no competing interests exist.

## Editor's evaluation

This fundamental work substantially advances our understanding of the role of Dlk1 in pituitary gland size and implicating WNT pathway. The evidence supporting the conclusions is compelling, with rigorous mouse genetic models and state-of-the-art ChipSeq and scRNA seq methods. The work will be of broad interest to cell biologists, developmental biologists and neuroendocrinologists.

## Introduction

How organ size is determined and maintained throughout life is a key question in biology. This process is complex and depends on the balance of proliferation/differentiation and apoptosis during development, during which the eventual size is attained. Following this, stem cells must repopulate the differentiated cells to maintain a stereotypical number. In some cases, organ size and cell composition can change dependent upon the life-cycle stage of the organism, or in response to the environment. The mammalian anterior pituitary gland (AP) is a master endocrine organ that integrates hypothalamic and peripheral cues to drive the release of circulating hormones. Life events such as stress, puberty, and

pregnancy-lactation cause remodelling of the AP to allow appropriate levels of hormone production, and changes in cell number and gland size (*Perez-Castro et al., 2012*). Since the stem cell capacity of SRY-related HMG-box 2 expressing (SOX2+) tissue-resident pituitary stem cells (PSCs) was demonstrated (*Andoniadou et al., 2013*; *Rizzoti et al., 2013*), many teams have integrated knowledge of developmental signalling pathways with information about their determination and maintenance (reviewed [*Russell et al., 2018*]). Importantly, the SOX2+ compartment acts not only as a source of cellular progenitors, but also in postnatal life directs the expansion of more committed cells by the production of paracrine signals, including WNT ligands (*Russell et al., 2021*). Due to these recent advances, the AP is an excellent system in which to study the fundamental process of organ size determination and homeostasis.

Imprinted genes are key determinants of body size in mammals, acting during early life to modulate growth and differentiation pathways (reviewed in [*Tucci et al., 2019*]). They represent ~100 transcripts in mammalian species that are epigenetically regulated such that only a single parental copy is expressed and the other silenced by a mechanism utilising DNA methylation. Imprinted gene functions converge on a small number of biological processes – neurobiology, placentation, and growth (reviewed in [*Tucci et al., 2019*]). We and others have shown that maintaining imprinted gene expression dosage within a narrow range is key to attaining the balance of growth to organ maturation (*Charalambous et al., 2012*; *Plagge et al., 2004*; *Tsai et al., 2002*). We recently noted that imprinted gene expression is enriched in the developing and postnatal pituitary gland, and we proposed that mis-regulation of gene dosage may underlie many of the endocrine features of human imprinting disorders including Silver Russell and Prader Willi Syndromes (*Scagliotti et al., 2021*). Importantly, a previously identified co-regulated subset of these genes, known as the imprinted gene network (IGN, [*Varrault et al., 2006*]), are co-expressed and represent some of the most abundant transcripts in the developing pituitary gland (*Scagliotti et al., 2021*).

Delta-like homologue 1 (*Dlk1*), a paternally expressed imprinted gene on mouse chromosome 12 (*Schmidt et al., 2000*; *Takada et al., 2000*), is a member of the IGN. The syntenic area on human chromosome 14 is the critical region for Temple Syndrome, and loss of *DLK1* expression is thought to cause phenotypes associated with this disorder; pre-and postnatal growth restriction with increased truncal obesity and precocious puberty (*Ioannides et al., 2014*). Alternative splicing of *Dlk1* results in several protein variants that have important functional differences. Full-length *Dlk1* encodes an ~60 kD protein that may be cleaved by the TACE protease ADAM17 at the extracellular cell surface to release soluble DLK1 into the circulation. Alternative splicing events skip the ADAM17 recognition site, resulting in a transmembrane form of DLK1 which cannot be cleaved (*Smas et al., 1997*). The signalling pathway by which DLK1 acts has yet to be elucidated. *Dlk1* encodes a protein with high sequence similarity to the NOTCH ligand Delta, but it lacks the DSL domain critically required for NOTCH interaction (*Smas and Sul, 1993*).

DLK1 is expressed in several progenitor cell populations where its expression is associated with the self-renewal to differentiation transition (reviewed in [*Sul, 2009*]). For example, addition of soluble DLK1 to cultured preadipocytes results in failure of these cells to differentiate into adipocytes whereas deletion of this gene causes premature differentiation (*Smas and Sul, 1993*). In the context of the postnatal neurogenic niche, the expression dosage of both membrane-bound and secreted DLK1 regulates the rate of self-renewal of the SOX2+ neural stem cells (*Ferrón et al., 2011*).

In mice, *Dlk1* expression is first detected from mid-gestation in a variety of mesodermal and neuroendocrine cell types, including the developing pituitary gland (*da Rocha et al., 2007*). After birth *Dlk1* expression is rapidly reduced, serum levels fall, and expression becomes restricted to a limited number of cell types including some neurons, adrenal and AP cells (*Sul, 2009*). In adult humans, *DLK1* is expressed in Growth Hormone (GH)-producing somatotrophs (*Larsen et al., 1996*). We and others have shown that *Dlk1* knockout mice are small with reduced GH production (*Puertas-Avendaño et al., 2011*; *Cheung et al., 2013*; *Cleaton et al., 2016*). However, *Gh* mRNA is not reduced following a conditional ablation of *Dlk1* in mature somatotrophs (*Appelbe et al., 2013*). We previously demonstrated that serum GH levels are elevated in mice that overexpress *Dlk1* from endogenous control elements at 6 months of age and that their GH levels fall less markedly following high-fat diet feeding (*Charalambous et al., 2014*). In contrast, pregnant dams lacking circulating DLK1 in pregnancy have a blunted increase in pregnancy-associated GH production (*Cleaton et al., 2016*). These data led us to hypothesise that DLK1 dosage modulates the size

of the GH reserve at critical life stages, and prompted us to investigate the molecular mechanism further.

Here, we manipulate *Dlk1* gene dosage in mice using both knock out and overexpression models. We show that *Dlk1* dosage regulates AP size independently of whole body weight, suggesting an autocrine/paracrine role for the product of this gene. Loss of *Dlk1* function leads to reduced AP volume by acting in a discrete developmental window to shift the balance of stem cell replication/commitment, by mediating the sensitivity of progenitor cells to the WNT pathway. Finally, increased *Dlk1* expression dosage increases the SOX2+ stem cell compartment, influencing organ expansion throughout the life course and increases the rate of postnatal replication in committed cells. This indicates that DLK1 may act postnatally to mediate paracrine signals between the stem and committed cell compartment of the AP. Overall, we conclude that *Dlk1* dosage determines pituitary size and postnatal stem cell homeostasis.

## Results

### Increasing the expression dosage of Dlk1 causes pituitary volume expansion without tumorigenesis

We previously demonstrated that adult GH was elevated in a transgenic model where extra copies of the *Dlk1* gene are introduced in the context of a 70 kb bacterial artificial chromosome (BAC, TG[Dlk1-70C]; *Charalambous et al., 2014*). This transgene drives gene expression in a similar temporal and tissue-specific pattern to the endogenous gene, but imprinting is not maintained due to lack of inclusion of the imprinting control region for the chromosome 12 cluster (*da Rocha et al., 2009*, *Figure 1A*). In the pituitary gland at 12 weeks of age, TG[Dlk1-70C] hemizygous mice (hereafter WT-TG) had increased *Dlk1* mRNA expression two- to threefold compared to wild-type (WT, *Figure 1B*). This resulted in elevated protein expression of all of the major isoforms of DLK1 (*Figure 1C*). In the adult AP, DLK1 was expressed in a subpopulation of all hormone-producing cells except corticotrophs (*Figure 1D*). The transgenic model did not modify the proportion of cells in the population that were DLK1+ (WT 36.5 ± 0.9%; WT-TG 36.3 ± 0.8%), rather, the same fraction of cells produced more protein (*Figure 1C*). Immunohistochemical staining for DLK1 clearly indicated that a proportion of the protein is membrane-localised (*Figure 1D*).

To investigate the physiological basis of increased GH production, we conducted a stereological analysis of the AP at 12 weeks in WT-TG mice and their WT littermates. Pituitary size was increased following *Dlk1* overexpression (*Figure 1E and F*), but the proportions of hormone-producing cells were not affected (*Figure 1D* and *Supplementary file 1*). We saw no evidence of pituitary tumours in any animal examined up to 1 year of age. Overall, this resulted in a 38% expansion in the somatotroph population, potentially increasing the GH-secretory reserve in WT-TG animals (*Figure 1G*). Thus, increasing the expression dosage of *Dlk1* causes pituitary volume expansion without a shift in hormone-producing cell fate. These data suggested that *Dlk1* might be acting during pituitary development to modulate AP size.

### Modulation of DLK1 gene dosage in the developing pituitary gland

In order to understand how *Dlk1* regulates pituitary size we generated embryos where the gene expression is increased (WT-TG), ablated (deletion of *Dlk1* from the active paternal allele, PAT) or ablated but expressing the transgene (PAT-TG) from matched litters (*Figure 2A*). In the whole pituitary gland at E18.5 *Dlk1* splice forms expressed from the TG[Dlk1-70C] transgene appeared similar in type and abundance to those expressed from the endogenous locus (*Figure 2B and C*).

*Dlk1* is expressed from early development of the AP, from at least E9.5 in Rathke's pouch (RP) and the overlying ventral diencephalon (*Figure 2D*). As development proceeds, *Dlk1* levels remain high both in progenitor cells of the cleft and lineage committing cells of the parenchyma (*Figure 2D*, WT and WT-TG). *Dlk1* expression could not be detected in PAT embryos at any stage, confirming that genomic imprinting is maintained in this tissue. Surprisingly, when expression from the TG[Dlk1-70C] transgene was examined in the absence of the endogenous allele, we could not detect *Dlk1* in the early development of the AP or midbrain. Rather, *Dlk1* expression was first detected in parenchymal cells at E15.5 of PAT-TG animals. In contrast, other tissues such as cartilage demonstrated robust expression of *Dlk1* in a pattern similar to that of the endogenous gene (*Figure 2D*). DLK1 protein expression mirrored that of the mRNA

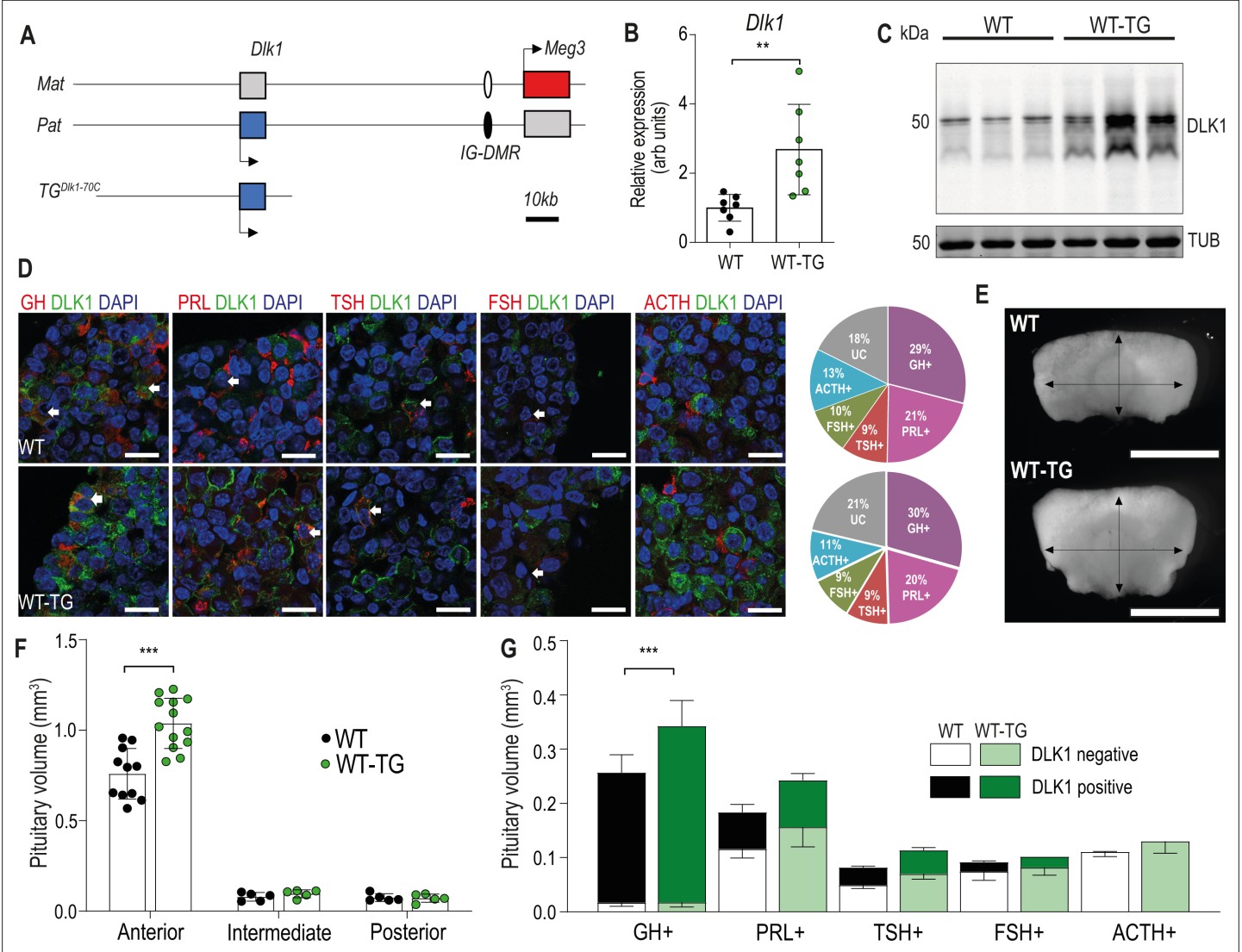

**Figure 1.** Increasing the expression dosage of *Dlk1* causes pituitary hyperplasia. (**A**) Schematic of part of the imprinted region on mouse chromosome 12 that contains the *Dlk1* and *Meg3* genes. *Dlk1* is expressed (blue box) from the paternally-inherited chromosome, *Meg3* from the maternally-inherited chromosome (red box). This expression pattern is established and maintained by an intergenic differentially methylated region (IG-DMR) which is paternally methylated (black oval) and maternally unmethylated (white oval). The *TG*$^{Dlk1-70C}$ transgene used in this study contains the whole *Dlk1* gene within ~70 kb of surrounding DNA, but does not contain the IG-DMR and is not imprinted. (**B**) RT-qPCR for *Dlk1* in whole pituitary from 12-week-old female hemizygous *TG*$^{Dlk1-70C}$ transgenic (WT-TG) mice and wild-type (WT) littermates. *Dlk1* expression is 2.7 x higher in transgenic animals, **p<0.01 compared by Mann-Whitney U test, n=7 per genotype. Bar shows mean +/-SD. (**C**) Western blots of extracts from whole pituitary of 12-week female WT and WT-TG mice. Alpha tubulin (TUB) is used as a loading control. Full-length and membrane bound isoforms at 50–60 kDa. (**D**) Fluorescence immunohistochemistry for DLK1 and pituitary hormones at 12 week. DLK1 in the adult AP is detected in all hormone-producing cells except corticotrophs; GH = somatotrophs, PRL = lactotrophs, TSH = thyrotrophs, FSH = gonadotrophs; ACTH = corticotrophs. Scale bar = 20 µm. Note some membrane localisation of DLK1. White arrows indicate co-expression. Proportion of hormone-labelled cells was quantified between mice of each genotype and shown as a pie chart on the right, UC = unclassified cell type. (**E**) Whole pituitary glands from adult WT-TG mice appear larger than those from WT littermates. Light field image of 12-week female glands, scale bar = 1 mm. (**F**) Anterior pituitary volume, but not intermediate lobe or posterior pituitary volume is increased in WT-TG animals compared to WT littermates. Twelve-week-old females, n=12/13 animals per genotype, compared by two-way ANOVA with Sidak's post-hoc multiple comparison test, *** p<0.001. (**G**) Overall cell proportion was not changed but absolute volume of hormone-producing cells was altered by *Dlk1* dosage (data shown in *Supplementary file 1a*). Groups are compared by two-way ANOVA and differ significantly according to genotype (p=0.0009); genotypes are compared for each cell type Sidak's multiple comparison test – WT and WT-TG animals have a significantly different proportion of GH-producing cells (*** p<0.001). (**F**) and (**G**) bars show mean values +/-SD.

The online version of this article includes the following source data for figure 1:

**Source data 1.** Zipped Excel file containing raw data used to generate graphs in *Figure 1*.

**Source data 2.** Zipped file containing source data for *Figure 1C* – gels with cropped bands highlighted and original gel images.

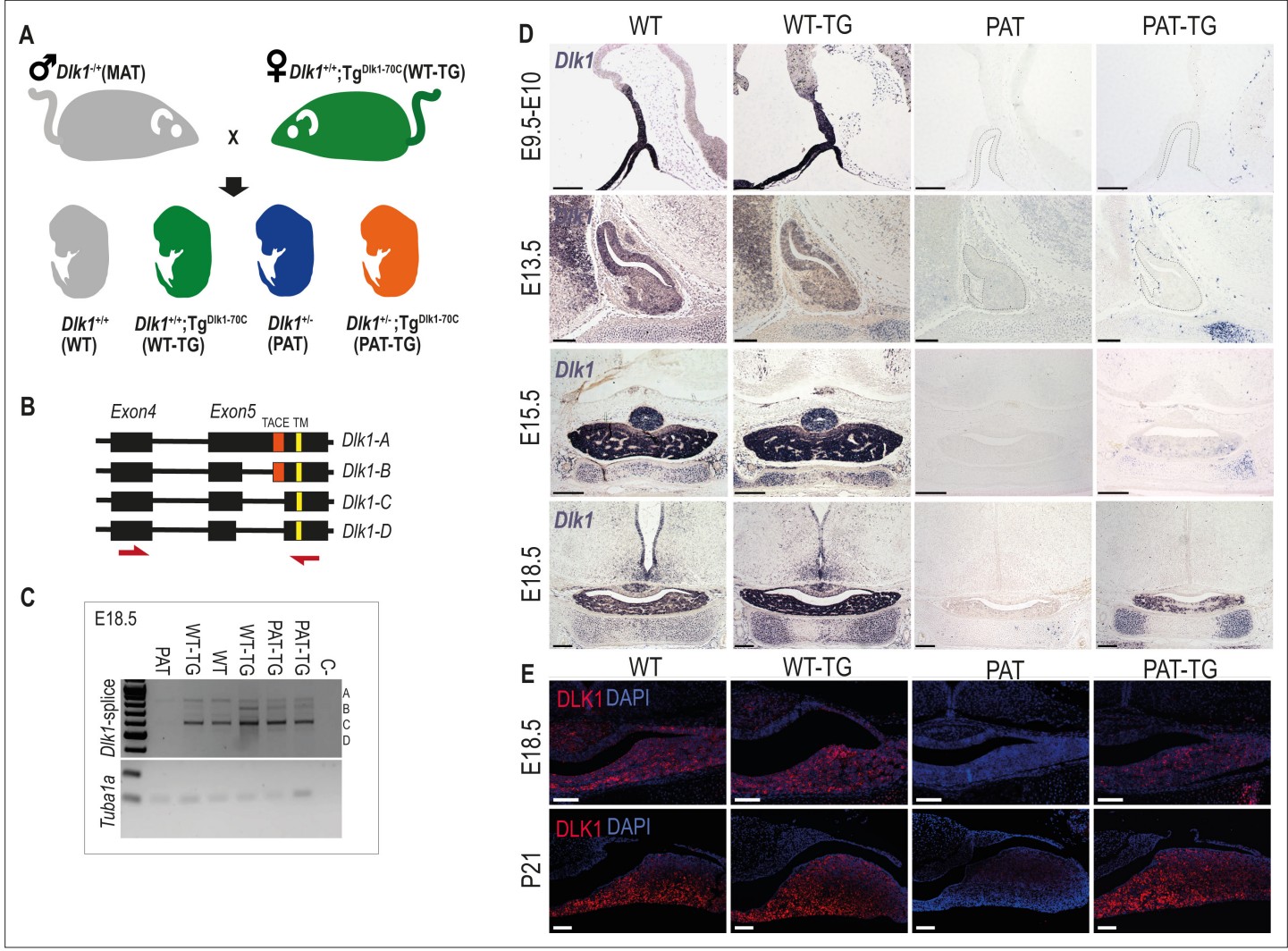

**Figure 2.** *Dlk1* imprinting and expression in the developing pituitary gland from the endogenous locus and *TG^Dlk1-70C* transgene. (**A**) Cross used to generate embryos and postnatal animals in the study. Males inheriting the deleted allele from the mother (maternal *Dlk1*^tm1Srbpa/+ heterozygotes or MATs) were crossed to females hemizygous for the *TG^Dlk1-70C* transgene (WT-TG), generating 4 genotypes, WT, WT-TG, paternal *Dlk1* ^+/tm1Srbpa heterozygotes (PATs) and mice inheriting a deleted paternal allele and the transgene (PAT-TG). (**B**) Schematic showing the known splice variants of *Dlk1*, A-D. Splicing occurs internally in exon 5 of the *Dlk1* gene. *Dlk1*-A and B retain an extracellular cleavage domain (TACE), in Dlk1-C and D this region is spliced out. All versions contain a single pass transmembrane domain (TM). Red arrows indicate location of primers used in (**C**). (**C**) Semi-quantitative PCR on embryonic day (**E**) 18.5 whole pituitary glands from the 4 genotypes shown in (**A**). Top – primers amplify the exon 4–5 region of *Dlk1* and can distinguish splice variants based on size. Bottom – alpha-tubulin (*Tuba1a*) was amplified as a loading control on each sample. (**D**) In-situ hybridisation for *Dlk1* in the developing pituitary gland from E9.5 to E18.5 in the 4 genotypes shown in (**A**). *Dlk1* expression is indicated by purple staining. Scale bars show 100 μm (E9.5 and E13.5, sagittal sections) and 200 μm (E15.5 and E18.5, frontal sections). (**E**) Immunohistochemistry (IHC) for DLK1 on frontal sections at E18.5 and postnatal day 21 (**P21**), counterstained with DAPI. Scale bars = 50 μm.

The online version of this article includes the following source data for figure 2:

**Source data 1.** Zipped file containing the source data for *Figure 2D* -gels with cropped areas highlighted and original gel images.

at E18.5 (*Figure 2E*) and at postnatal day 21 (P21) a similar pattern was observed. These data indicate that endogenous *Dlk1* is expressed from the onset of pituitary development, in both progenitor and lineage committing cells, and that some regulatory sequences necessary for the full repertoire of *Dlk1* gene expression are not located within the 70 kb genomic region delineated by the transgene.

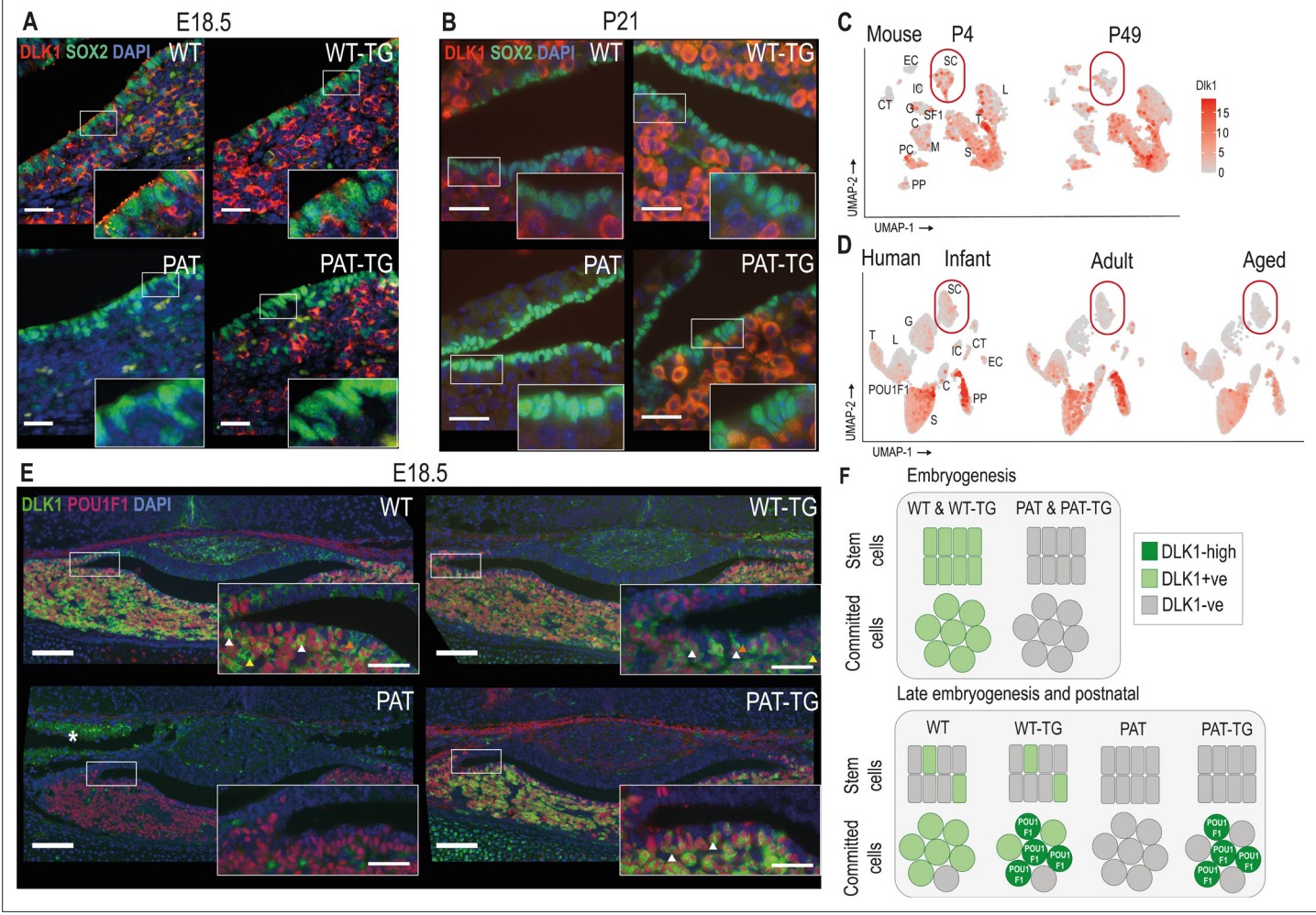

**Figure 3.** DLK1 expression in the pituitary gland is dynamically regulated in two distinct compartments, only one of which is recapitulated by the *TG^{Dlk1-70C}* transgene. (**A**) IF of DLK1 and SOX2 in the E18.5 pituitary. SOX2 expression is high in the epithelial cells lining the pituitary lumen, as expected. In WT and WT-TG mice there is co-expression of DLK1 and SOX2 (inset), as well as high levels of DLK1 expression in the SOX2-negative parenchyma. In the PAT-TG pituitary DLK1 expression is not detected in the SOX2-positive compartment. Scale bars = 25 µm. (**B**) IF of DLK1 and SOX2 in the P21 pituitary. The majority of DLK1 expression is outside the SOX2-positive compartment of all genotypes. Scale bars = 25 µm. (**C**) UMAP plot illustrating expression of *Dlk1* in sc-RNAseq at P4 (*Cheung and Camper, 2020*) and P49 mice (*Cheung et al., 2018*) in the postnatal mouse pituitary, indicating high expression in the POU1F1 lineage (S somatotrophs, T thyrotrophes, L lactotrophs), with additional expression in a subset of *Sox2*-positive stem cells (SC), outlined by red box. EC endothelial cells, IC immune cells, CT connective tissue, G gonadotrophs, SF1 steroidal factor 1 progenitors, C corticotrophs, M melanotrophs, PC proliferating cells, PP posterior pituitary. (**D**) UMAP plot illustrating expression of *DLK1* in sn-RNAseq from infant, adult and aged human pituitary gland (*Zhang et al., 2022*), indicating high expression in the POU1F1 lineage (S somatotrophs, T thyrotrophs, L lactotrophs), with additional expression in a subset of *SOX2*-positive stem cells (SC). Additional cell labels as in (**C**). (**E**) IF of DLK1 and POU1F in the E18.5 pituitary. DLK1 and POU1F1 are highly co-expressed in all *Dlk1*-expressing genotypes (white arrows). In WT and WT-TG there is additional DLK1 expression in marginal zone cells (orange arrow) and parenchymal cells (yellow arrow) that are POU1F1-negative. This expression is absent from the PAT-TG. Scale bar = 100 µm, inset 25 µm. * indicates background autofluorescence from blood cells. (**F**) Summary of *Dlk1* expression across embryonic and early postnatal development in the mouse, with contribution from the transgene.

The online version of this article includes the following figure supplement(s) for figure 3:

**Figure supplement 1.** *Dlk1* may be a direct transcriptional POU1F1 target.

## Dlk1 exhibits complex temporal and spatial regulation in the AP which is only partially recapitulated by the TG^{Dlk1-70C} transgene

Since DLK1 expression in the pituitary progenitor/stem population has not previously been described in detail, we examined its temporal co-expression with the stem cell marker SOX2. In late embryogenesis and postnatally SOX2+ PSCs occupy the periluminal marginal zone (MZ) of the pituitary

gland. In WT and WT-TG E18.5 embryos, DLK1 is clearly co-expressed with SOX2 in this compartment (*Figure 3A*). However, by postnatal day 21 (P21), DLK1+ cells are largely absent from the MZ of animals with an intact *Dlk1* gene (*Figure 3B*). As expected, deletion of *Dlk1* from the paternal allele ablates all protein expression. However, expression from the TG^Dlk1-70C transgene was absent from the MZ at both stages.

We recently reanalysed publicly-available single-cell sequencing (scRNA-seq) data of the mouse pituitary gland generated by the Camper lab (*Cheung et al., 2018*), (*Cheung and Camper, 2020*) in order to catalogue imprinted gene expression in the early postnatal (P4) and adult (P49) gland (*Scagliotti et al., 2021*). These data confirmed our immunohistochemical analyses (*Figure 1D*) of enriched expression of *Dlk1* in somatotrophs, lactotrophs and thyrotrophs, as well as in a subset of *Sox2*-expressing stem cells (*Figure 3C*). The scRNA-seq data also appeared to confirm the reduction in *Dlk1* gene expression in the stem cell compartment between early and late postnatal stages (17.6% at P4 to 14.8% at P49, *Figure 3C*). Similarly, the proportion of cells expressing *DLK1* in the human postnatal PSC compartment decreases from youth to adulthood as shown by analyses on human snRNA-seq pituitary datasets (*Zhang et al., 2022*; 14.1% in infant, 5.7% in adult and 8.9% in aged, *Figure 3D*). As in the mouse, *DLK1* expression in the human pituitary is most abundant in somatotrophs, lactotrophs and thyrotrophs. These three cell types are derived from a common embryonic lineage progenitor expressing the transcription factor POU1F1, and the mature hormonal lineages continue to express this transcription factor to directly regulate hormonal genes (*Camper et al., 1990*; *Li et al., 1990*). We found a high level of co-expression between DLK1 and POU1F1 in the WT and WT-TG pituitary gland at E18.5, though DLK1+ cells were also clearly visible in the MZ and in POU1F1- parenchymal cells (*Figure 3E*). In contrast, in the PAT-TG pituitary, DLK1 expression was confined solely to POU1F1+ cells. These data indicate that the TG^Dlk1-70C transgene contains *cis* regulatory sequences necessary for *Dlk1* expression in POU1F1+ cells but not those required for expression in stem cells and other hormonal lineages (*Figure 3F*). To determine whether *Dlk1* might be a direct target of POU1F1 we explored a POU1F1 ChIP-seq dataset from GH-expressing cells derived from rats (*Skowronska-Krawczyk et al., 2014*). By mapping the rat POU1F1 binding sites onto the mouse genome we identified three potential POU1F1 binding sites within 120 kb of the *Dlk1* gene (*Figure 3—figure supplement 1*). These sites overlapped other genomic features indicative of regulatory activity including evolutionary conservation, CpG islands and previously mapped histone modifications associated with enhancer activity in the embryonic brain (*Gorkin et al., 2020*). One of these binding sites was localised within the 70 kb included in the TG^Dlk1-70C transgene. We suggest that *Dlk1* is likely to be a direct POU1F1 target in both the WT and TG^Dlk1-70C transgenic mice.

## Dlk1 dosage modulation controls AP volume in two distinct developmental periods and independently of body size

We next explored the consequences of *Dlk1* dosage modulation on pituitary size by performing stereological volume measurements on a time course of embryonic and postnatal pituitaries. First addressing whole organism weight, we calculated total body weight of each genotype relative to WT littermates. PAT animals were growth restricted (~85% WT weight at E18.5) in late gestation, as previously reported (*Cleaton et al., 2016*). WT-TG mice were not significantly altered in mass, but the TG^Dlk1-70C transgene rescued some of the growth deficit on a *Dlk1*-deleted background; PAT-TG animals weighed ~92% WT mass at E18.5 (*Figure 4A*, and *Supplementary file 1b*). Postnatally PAT animals were further growth restricted, reaching a minimum proportion of WT weight at P14 (~55%), after which the animals commenced catch-up growth, reaching 82% WT weight by weaning at P21.

Next, we used stereological methods to measure pituitary volume and compared relative volumes by genotype. In contrast to body weight, the volume of the AP is already significantly reduced in size at E13.5 (61% WT), prior to embryonic growth restriction. As predicted by the lack of *Dlk1* expression from the TG^Dlk1-70C transgene at this stage (*Figure 2B*), the WT vs WT-TG and PAT vs PAT-TG are indistinguishable in volume (*Figure 4B and C*, *Supplementary file 1c and d*). Subsequently, the *Dlk1*-ablated AP maintains this deficit in volume (50–65% WT volume) into postnatal life, with some evidence of volume catch-up at weaning. The TG^Dlk1-70C transgene is unable to consistently rescue the deficit in pituitary volume that is established during embryogenesis (PAT and PAT-TG pituitary volumes are broadly overlapping). In contrast, in WT-TG animals increased *Dlk1* expression from the TG^Dlk1-70C transgene did cause an increase in pituitary volume. This increase in volume was evident in

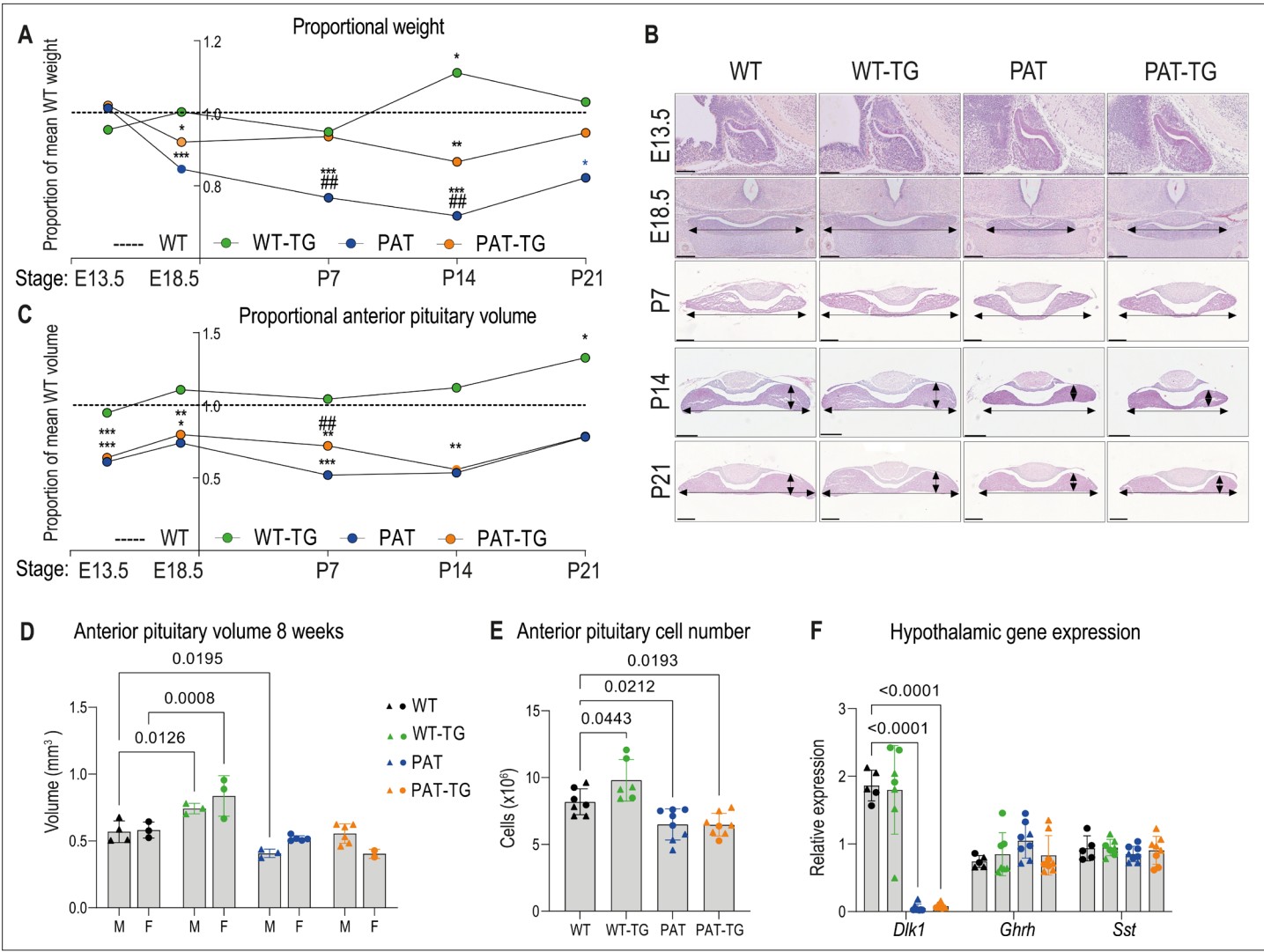

**Figure 4.** DLK1 dosage during embryogenesis and early life modulates anterior pituitary volume independent of whole animal mass. (**A**) Time-course of proportional body weight by genotype from E13.5 to P21, derived from data in **Supplementary file 1b**, sexes are combined since body weight is not sexually dimorphic at these stages. Lines and points show mean proportional weight of embryo/pup for WT-TG, PAT and PAT-TG animals compared to WT littermates (dotted line at 1.0). The unmodified weight data was compared using a One-Way ANOVA with Bonferroni's posthoc test comparing each genotype with WT (*p<0.05, ***p<0.001), and PAT with PAT-TG (## p<0.01). (**B**) Haemotoxylin and eosin-stained pituitary glands at E13.5 (sagittal view), E18.5, P7, and P21 (frontal view) across the four genotypes. Scale bars show 100 μm (E13.5), 200 μm (E18.5), (**P7**) and 300 μm (**P14, P21**). (**C**) Time-course of proportional pituitary volume by genotype from E13.5 to P21, derived from data in **Supplementary file 1c and d**. Lines and points show mean volume of the anterior pituitary gland for WT-TG, PAT, and PAT-TG animals as a proportion of WT littermates (dotted line at 1.0). Genotypes were compared in unmodified volume data using a one-way ANOVA with Bonferroni's posthoc test comparing each genotype with WT (*p<0.05, **p<0.01, ***p<0.001), and PAT with PAT-TG (## p<0.01). (**D**) Anterior pituitary volumes of male triangles, (**M**) and female circles, (**F**) mice at postnatal week 8. Volumes were compared by sex and genotype with a two-way ANOVA and within genotypes using a Dunnett's multiple comparison test. Volumes were significantly different by genotype (p<0.0001), and p values of significant post-hoc comparisons are shown on the graph, n = 2-6/sex/genotype. (**E**) Estimated total cell numbers of APs from (**D**). Sexes were combined and genotypes compared using One-Way ANOVA (p<0.0001 overall). Each genotype (n = 6-8 per genotype)was compared to WT using Dunnett's multiple comparison test and p values shown on the graph. (**D, E**) Source data shown in **Supplementary file 1e**. (**F**) RT-qPCR quantitation of gene expression in the hypothalamus at 8 weeks in samples from D, normalised to actin. For each gene, expression levels were compared between genotypes as in (**E**), post-hoc p values are shown on the graph. (**D–F**) Bars show the mean of the data +/-SD.

The online version of this article includes the following source data and figure supplement(s) for figure 4:

**Source data 1.** Zipped file containing raw data used to generate graphs in **Figure 4**.

**Figure supplement 1.** *Dlk1* dosage modulates hormone levels and early life endocrine physiology.

the second postnatal week and became statistically significant at P21 (133% WT volume, consistent with 38% increased AP mass as adults, *Figures 3C and 1F*). Adjacent structures, the posterior pituitary and postnatal intermediate lobe volume did not differ between genotypes (*Supplementary file 1c and d*) indicating that *Dlk1* acts selectively on pathways that regulate AP size. Moreover, AP volume expansion only develops when elevated *Dlk1* expression in the parenchyma coexists with with a *Dlk1* +stem cell compartment.

Finally, we determined AP volume in a cohort of adult animals with both sexes represented (*Figure 4D*, and *Supplementary file 1e*). At 8 weeks of age, WT-TG male and female animals have reached ~140% WT volume, similar to that observed in the original 12-week-old female cohort (*Figure 1F*). Consistent with the volume catchup observed at 21D, PAT and PAT-TG AP volume was no longer significantly different from WT, except for in PAT males. Importantly, PAT-TG AP volume did not expand beyond that of the PAT, further supporting the finding that *Dlk1* is required in both stem and parenchymal compartments to drive increased volume. By stereological estimation of cell number in the adult anterior lobe, we found that WT-TG AP contained a larger total number of cells (*Figure 4E*), indicating that the expansion occurs by hyperplasia. PAT and PAT-TG anterior lobes contained fewer cells, suggesting a long-term impact of developmental size reduction.

Somatotroph and lactotroph volume expansion has previously been observed in mice following transgenic overproduction of growth hormone releasing hormone (*Ghrh*), (*Stefaneanu et al., 1989*). To test whether pituitary hyperplasia in WT-TG was secondary to alterations in hypothalamic neuropeptide production, we measured the expression of *Ghrh* and somatostatin (*Sst*) in the adult hypothalamus (*Figure 4F*). Consistent with the lack of TG^Dlk1-70C -derived *Dlk1* expression in the midbrain during embryogenesis (*Figure 2D*), *Dlk1* mRNA levels were similar between WT and WT-TG mice, and undetectable in both PAT and PAT-TG mice. Expression of *Ghrh* and *Sst* did not differ between the genotypes (*Figure 4F*).

## Dlk1 dosage modulation does not prevent hormone expression but modifies GH levels and pubertal physiology

We surveyed the expression of pituitary hormones at E18.5, when all the hormonal axes have been established. Loss or gain of *Dlk1* dosage did not cause any gross change in hormonal cell localisation or expression at this stage (*Figure 4—figure supplement 1A–Y*). To investigate if the increased AP volume in the WT-TG animals has any physiological consequences, we measured GH levels in female mice in the peri-pubertal period. In mice, GH levels rise around weaning in the third postnatal week to reach adult levels by the fifth postnatal week (*Figure 4—figure supplement 1Z*). GH levels were elevated in young WT-TG mice and though the trend was detected later it became obscured by considerable variability (*Figure 4—figure supplement 1Z*). We could not detect a reduction in GH levels in PAT mice at any stage (data not shown). Temple Syndrome and *DLK1* mutations in humans are associated with precocious menarche (*Ioannides et al., 2014*; *Dauber et al., 2017*). Using day of vaginal opening (VO) as a proxy for onset of puberty we determined that, contrary to our expectations, increased expression of *Dlk1* resulted in earlier pubertal onset. However, deletion of *Dlk1* did not alter the timing of puberty, but females were relatively smaller when they entered this state (*Figure 4—figure supplement 1A*).

## Dlk1 regulates embryonic pituitary size by acting in a discrete developmental window to shift the balance of stem cell replication/commitment

*Dlk1* is expressed from the onset of pituitary morphogenesis in Rathke's pouch and the overlying ventral diencephalon (*Figure 2D*). Despite this early expression, we could detect no difference in the expression of transcription factors and signalling molecules that have previously been shown to regulate morphogenesis and growth of the gland at this stage (*Figure 5—figure supplement 1*). From ~E13.5, as pituitary progenitor cells exit the cell cycle and differentiate they move ventrally away from the periluminal cleft (*Davis et al., 2011*). Proliferating cells proximal to the lumen of the cleft are SOX2+. DLK1 expression is found in the SOX2+ proliferating cells in the cleft as well as in the newly differentiating cells (*Figure 3D*). Loss of *Dlk1* in PAT and PAT-TG animals caused a transient reduction in cell proliferation in the cleft at E13.5 (*Figure 5A–D*, *Supplementary file 1f*). There was a concomitant increase in the proportion of cells expressing the lineage marker POU1F1 (*Figure 5E and G*).

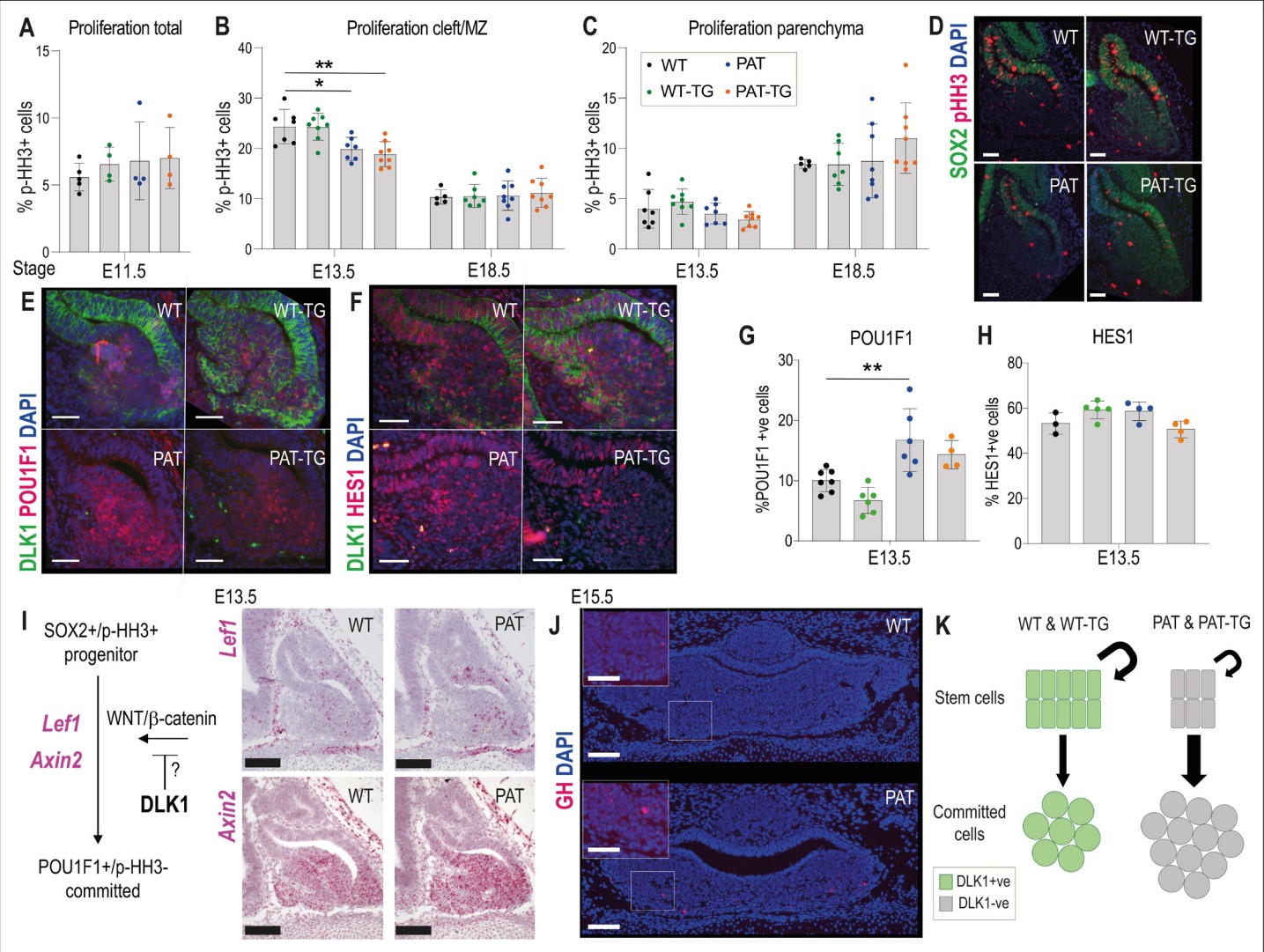

**Figure 5.** Loss of *Dlk1* expression in an embryonic time window shifts progenitor cells from proliferation to differentiation. (**A**) Proportion of phosphohistone H3 (pHH3)-positive cells to total cells in the pituitary at E11.5 (n = 4-5/genotype).(**B**) Proportion of pHH3-positive cells in the morphological stem cell compartment (cleft/marginal zone (MZ)) at E13.5 and E18.5 across the 4 *Dlk1* genotypes. (**C**) Proportion of pHH3-positive cells in the parenchymal region across the 4 *Dlk1* genotypes. (**B**) and (**C**) for each time point genotypes were compared using a one-way ANOVA with Bonferroni's posthoc test comparing each genotype with WT (*p<0.05, **p<0.01), and PAT with PAT-TG, (n = 5-8/genotype). (**D**) Representative image showing increased proliferation in the SOX2 +stem cell compartment of *Dlk1*-expressing (WT and WT-TG) embryos at E13.5 compared to non-expressing embryos (PAT and PAT-TG). (**E**) Representative image showing increased number of POU1F1 cells newly differentiating in the parenchyma of the *Dlk1* non-expressing E13.5 pituitary (PAT and PAT-TG), compared with the *Dlk1*-expressing (WT and WT-TG) gland. (**F**) Representative image showing no difference in the number of cells expressing nuclear HES1 in *Dlk1*-expressing (WT and WT-TG) embryos at E13.5 compared to non-expressing embryos (PAT and PAT-TG). (**D-F**) scale bar = 50 μm. (**G**) Proportion of POU1F1-positive cells in the parenchymal region at E13.5 across the 4 *Dlk1* genotypes. Genotypes were compared using a one-way ANOVA with Bonferroni's posthoc test comparing each genotype with WT (**p<0.01, n = 4-7/genotype). (**H**) Proportion of nuclear HES1-positive cells in the E13.5 pituitary of the 4 *Dlk1* genotypes. Genotypes were compared using a one-way ANOVA with Bonferroni's posthoc test comparing each genotype with WT (none significant, n = 3-5/genotype). (**I**) Left: schematic showing how *Dlk1* might interact with the WNT signalling pathway which promotes commitment towards the *Pou1f1* lineage. Right: RNA scope in-situ hybridisation for *Lef1* and *Axin2* in E13.5 pituitary from WT and PAT littermates. *Lef1* and *Axin2* expression is indicated by pink staining, counterstained in blue (haematoxylin). Scale bar = 100 μm. (**J**) Representative IF for GH at e15.5 in WT and PAT pituitary (coronal section). A small number of GH-positive cells were detected in 6/6 PAT compared to 0/4 WT animals. Scale bar = 100 μm or 50 μm (inset). (**K**) Schematic summarising the impact of loss of *Dlk1* expression in the embryonic pituitary gland.

The online version of this article includes the following figure supplement(s) for figure 5:

**Figure supplement 1.** Altered dosage of *Dlk1* does not affect expression of early morphological signalling genes in Rathke's pouch.

Several molecular signalling pathways have been proposed to regulate the emergence of the POU1F1 lineage including NOTCH (*Zhu et al., 2006*) and WNT pathways (*Potok et al., 2008*;, *Olson et al., 2006*). While nuclear HES1, a marker of active NOTCH signalling, was abundantly expressed in the AP at this timepoint, we did not observe a difference in the proportion of HES1+ nuclei between *Dlk1* genotypes (*Figure 5F and H*). However, we detected increased expression of *Lef1* and *Axin2*, downstream targets of the WNT/β-catenin pathway (*Hovanes et al., 2001*;, *Jho et al., 2002*; *Figure 5I*). Taken together, we propose that loss of *Dlk1* in PAT and PAT-TG animals reduces AP size by altering the balance of cells remaining in the replicating stem cell pool to those that commit to the *Pou1f1* lineage (*Figure 5J*). This balance may be shifted by the action of *Dlk1* on the WNT/β-catenin pathway (*Figure 5I*).

## Elevated DLK1 dosage causes increased proliferation of the postnatal AP only when expression is retained in the stem cell compartment

In postnatal life, overexpression of *Dlk1* from the TG$^{Dlk1-70C}$ transgene caused an increase in AP volume, but only when WT *Dlk1* expression was retained (compare WT-TG with PAT-TG, *Figure 3A*). These data suggested that DLK1 produced by marginal zone stem cells and parenchymal cells, might act in concert to promote postnatal proliferation. We consistently observed that WT-TG mice have elevated parenchymal proliferation at P14, the time of maximal expansion of the committed cells prior to puberty (*Taniguchi et al., 2001*). This increase in proliferation was not observed in the PAT-TG mice (*Figure 6A–C*). Further, the SOX2+ stem cell compartment is modified in WT-TG and not PAT-TG mice; WT-TG animals at weaning have an increased proportion of SOX2+ cells (*Figure 6D and F*) and their overall compartmental volume (*Figure 6E*) in both the MZ and parenchymal clusters is increased. To determine if this translated to an increase in active adult stem cells we performed a stem cell colony-forming assay in adult mice (*Andoniadou et al., 2012*). In this assay, the AP is dissociated and plated at low density in culture media that promotes only the survival of PSCs. The number of colonies after 7 days of culture is indicative of the proportion of active stem cells in the source gland (*Figure 6G*). In cultures from the AP of 12-week-old WT-TG mice, we observed an increase in colony forming units (CFUs) compared to WT littermates (*Figure 5H, I and J*), indicating that increasing *Dlk1* expression dosage increases the lifelong stem cell reserve.

## Discussion

We previously demonstrated that adult TG$^{Dlk1-70C}$ (WT-TG) mice produce more pituitary *Gh* mRNA, have elevated fasting GH levels and concomitant changes to whole-body lipid oxidation pathways (*Charalambous et al., 2014*). The increase in the GH reserve can be explained by data presented in this study, since adult TG$^{Dlk1-70C}$ mice have a hyperplastic pituitary gland associated with a~40% increase in the size of the somatotroph population. We did not observe an increase in linear growth in the WT-TG mice (*Figure 4A*). This may be explained by the timing of AP volume expansion, which occurs in the third postnatal week (*Figure 4B*). In mice, the majority of GH action on linear growth occurs prior to P21 (*Lupu et al., 2001*). While we did observe increased circulating GH in WT-TG animals at P21 (*Figure 4—figure supplement 1*), this may have been too late to promote the pre-pubertal growth spurt. Moreover, growth signalling by GH requires regulated pulsatile secretion of the hormone (*Huang et al., 2019*), generated by the homotypic somatotroph cell network (*Bonnefont et al., 2005*). Future work could investigate how DLK1 dosage-mediated alterations in cell number influences the integrity of this network. In contrast to a role in growth, we have observed that *Dlk1* dosage is elevated during life periods when enhanced peripheral lipid oxidation is beneficial for survival; during suckling (*Charalambous et al., 2012*) and in the mother during pregnancy (*Cleaton et al., 2016*). These are periods of life when lipogenesis may be inhibited and tissue fatty acid (FA) oxidation promoted to spare scarce glucose for growth. We speculate that imprinting may be acting on DLK1 dosage to modulate the GH axis and shift the metabolic mode of the organism toward peripheral FA oxidation and away from lipid storage. Therefore, disruption of DLK1 dosage has important consequences for energy homeostasis and metabolic disease.

*Dlk1* imprinting is maintained in several embryonic cell populations that take part in the formation of the mature anterior pituitary gland. The endogenous gene is expressed exclusively from the paternally-inherited allele in the ventral diencephalon and in developing Rathke's pouch at E10

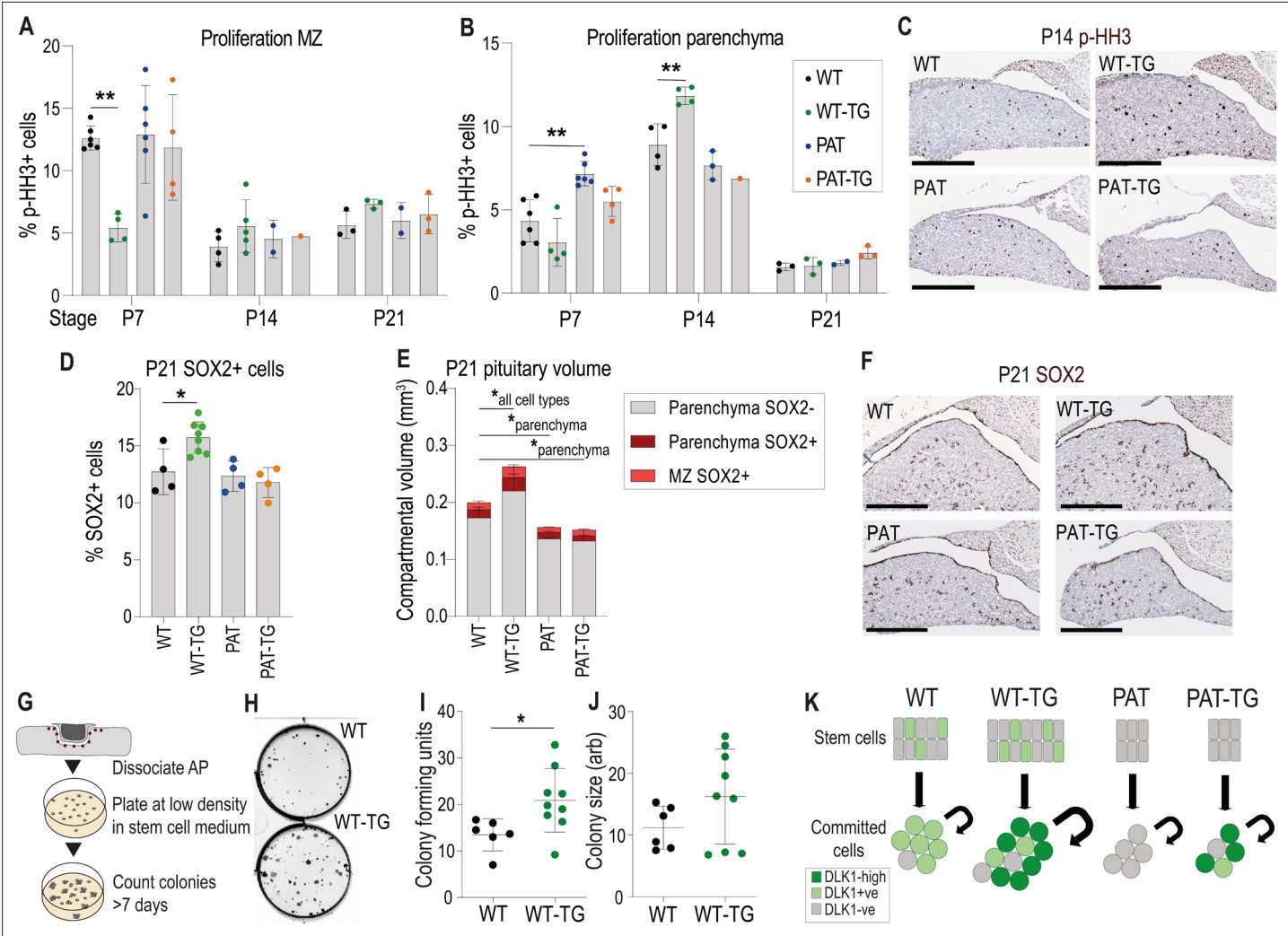

**Figure 6.** Increased proliferation and stem cell number occurs when Dlk1 dosage is elevated in both stem and parenchymal cells. (**A**) Proportion of phospho-histone H3 (**p–HH3**)-positive cells in the marginal zone (MZ) of the anterior pituitary at postnatal days 7, 14, and 21. (**B**) Proportion of phospho-histone H3 (**p–HH3**)-positive cells in the parenchymal zone (MZ) of the anterior pituitary at postnatal days 7, 14, and 21. (**A**) and (**B**) for each time point genotypes were compared using a one-way ANOVA with Bonferroni's posthoc test comparing each genotype with WT (**p<0.01, n = 1-6/ genotype). (**C**) Representative images of immunohistochemistry for p-HH3 at P14 (brown staining, blue counterstain with haemotoxylin), showing increased proportion of positive staining in the parenchyma of the WT-TG anterior pituitary. Scale bars = 250 μm. (**D**) Proportion of SOX2-positive cells in the anterior pituitary at postnatal day 21. Genotypes were compared using a one-way ANOVA with Bonferroni's posthoc test comparing each genotype with WT (*p<0.05, n = 4-8/genotype). (**E**) Volumes of P21 anterior pituitaries subdivided into categories dependent on location and SOX2 staining. Genotypes were compared using a two-way ANOVA and differ significantly according to genotype (p<0.0002), category (p<0.0001) and the interaction between them (p<0.0001). Genotypes are compared in each category to WT using Dunnett's multiple comparison test. WT-TG animals have a significantly increased cell volume in all categories, whereas PAT and PAT-TG animals differ only in parenchymal cell volume (*p<0.05). Bars show mean +/-SD, n=4–8 animals/genotype. (**F**) Representative images of immunohistochemistry for SOX2 at P21 (brown staining, blue counterstain with haemotoxylin), showing increased proportion of positive staining in the WT-TG anterior pituitary. Scale bars = 250 μm. (**G**) Methodology for determining the number of tissue-resident stem cells in the AP. Whole pituitary is dissected and the posterior lobe removed. The tissue is enzymatically dissociated and counted. A fixed number of cells are seeded at low density into culture media that promotes stem cell growth. After 7 days colonies are counted. Each colony represents a stem cell from the original organ. (**H**) Image of pituitary stem cell cultures stained after growth for 7 days, derived from WT (top) and WT-TG (bottom) adult animals. Black dots in the plate indicate colonies derived from a single colony forming unit (CFUs). (**I**) Number of CFUs from pituitary stem cells following 7 days of culture on 4 separate occasions from a total of n=6 WT and n=9 WT-TG, 13–16 week adult animals. (**J**) Area of CFUs from the stem cell cultures in (**I**). Genotypes were compared using a Student's t-test, *p<0.05. (**K**) Schematic summarising the action of *Dlk1* in the different compartments of the AP.

(*Figure 2D*), then in the SOX2+ progenitor populations in mid-late gestation (*Figure 3A*). As development proceeds, the proportion of DLK1+ SOX2+ cells decreases to approximately 10% of all stem cells in the mature gland (*Figure 3B–D*). Concomitantly, expression of *Dlk1* initiates in committed cells of the parenchyma from their appearance at ~E13.5 (*Figure 2D*), and is maintained in hormone producing cells, being particularly abundant in cells of the POU1F1 lineage, somatotrophs (GH+/DLK1+=93% of all GH+), lactotrophs (PRL+/DLK1+=37% of all PRL+), and thyrotrophs (TSH+/DLK1+=40% of all TSH+, *Supplementary file 1a*). Indeed, DLK1 shares considerable overlap with POU1F1 in late gestation and may be a direct target of this transcription factor, since three experimentally validated POU1F1 binding sites are located within 120 kb of the *Dlk1* gene (*Figure 3—figure supplement 1*), (*Skowronska-Krawczyk et al., 2014*). Additional experiments, such as measurement of *Dlk1* levels in *Pou1f1*-deleted mice, or targeted deletion of the putative POU1F1 binding sites at the *Dlk1* locus would be required to definitively establish this finding. *Dlk1* may be part of an autoregulatory feedback loop since, at least in vitro, expression of *Dlk1* represses the action of POU1F1 on the *Gh* promoter (*Ansell et al., 2007*). The expression pattern of *Dlk1* in the progenitor compartment mirrors that of other imprinted genes in the 'Imprinted Growth Network, IGN', such as *Igf2*, *Cdkn1c*, and *Grb10* (*Scagliotti et al., 2021*). These genes have been suggested to comprise the hub of a core pathway that regulates embryonic growth and is progressively deactivated in the perinatal period (*Lui et al., 2008*; *Varrault et al., 2006*). The retention of *Dlk1* expression in a small proportion of SOX2+ stem cells in adults suggests heterogeneity of this population, and that a small number of cells retain an embryonic progenitor cell-like phenotype. Well powered analysis of single-cell sequencing datasets of the PSC population over the life course will be required to validate this assertion.

We observed a profound reduction in the volume of the anterior pituitary gland (40–50%) of *Dlk1*-deficient animals at E13.5, before we observed any change in overall body weight (*Figure 3B*). *Dlk1* is expressed at very high levels in the developing AP compared to other embryonic tissues (compare for example the AP expression with cartilage at E15.5 in *Figure 2D*). The relative volume of the *Dlk1*-deficient AP is maintained at ~40–50% WT levels until the third postnatal week, when there appears to be some catch-up in volume. Importantly and in contrast to the total body weight, addition of the TG[Dlk1-70C] transgene does not consistently rescue pituitary volume in *Dlk1*-deficient mice (*Figure 3B*). These data indicate that loss of *Dlk1* in the SOX2+ progenitors in the early RP causes a persistent reduction to AP volume. Consistently, we observed a transient reduction in proliferation of the RP progenitor population at E13.5. This was associated with an increase in the size of the nascent POU1F1+population, cells which have recently left the dorsal proliferative zone and started along the lineage commitment pathway (*Figure 5E and G*). Consistently, the expression of a terminal hormonal marker of the POU1F1 lineage, GH, was observed earlier in *Dlk1*-deficient animals (*Figure 5J*). SOX2+ progenitor cell cycle exit is regulated by multiple signalling pathways including the NOTCH and WNT/β-catenin pathways. HES1 is a major downstream transcriptional target of NOTCH in the developing pituitary gland (*Goto et al., 2015*). Despite robust nuclear expression of HES1 in the RP at E13.5, we saw no difference in the proportion of HES1+ cells between genotypes, suggesting that NOTCH signalling perturbation at this stage is not responsible for pituitary volume reduction in *Dlk1*-deleted embryos. Whether DLK1 is a direct inhibitor of NOTCH signalling is still unresolved, since while some have reported biochemical interactions between DLK1 and NOTCH1 (*Baladrón et al., 2005*), others have failed to co-immunoprecipitate the two proteins in a biologically relevant context (*Wang et al., 2010*).

We observed that the expression downstream targets of WNT/β-catenin signalling, *Lef1* and *Axin2*, were increased in expression domain and intensity in PATs. Exit from the proliferating progenitor pool for commitment to the POU1F1 lineage is regulated by an interplay between the transcription factor Prophet of PIT1 (PROP1) and β-catenin binding to and activating the POU1F1 promoter (*Olson et al., 2006*). Increased activation of β-catenin targets in *Dlk1*-depleted embryos suggests that the normal function of DLK1 may be to act to either reduce WNT production or reduce the sensitivity of cells to the WNT pathway.

The expression of *Dlk1* in both the stem cell and mature hormone-producing compartment of the pituitary gland is perplexing, since it suggests an interaction between DLK1 expression in the progenitor compartment and in the niche. A precedent for this lies in the postnatal neuronal subventricular zone, where DLK1 expression is required in both the stem cells and in niche astrocytes to maintain adult stem cell potency (*Ferrón et al., 2011*). Here, we observed that the postnatal WT-TG AP was

expanded in volume compared to WT littermates, driven by increased proliferation of the parenchymal cells in the second postnatal week. A similar volume expansion was not observed when PAT-TG animals were compared to PAT littermates (*Figures 4C and 6B*). These data suggest that excess *Dlk1* can promote pituitary hypertrophy, but only if DLK1 is expressed in the SOX2+ compartment when dosage is increased in the parenchyma. Intriguingly, the increase in cell division mediated by excess DLK1 at P14 is in the parenchyma, and therefore DLK1 is not acting cell autonomously in stem cells to promote proliferation, yet SOX2+ DLK1+ cells are required for volume expansion. This suggests that the SOX2+ DLK1+ cells may produce a signal that promotes parenchymal proliferation, analogous to previous data where we demonstrated that the postnatal stem cells promote committed cell proliferation by a WNT-mediated pathway (*Russell et al., 2021*). We speculate that DLK1 production from stem cells might comprise part of this paracrine signalling system that is required for the size regulation of the mature pituitary gland. In concert, increased expression of DLK1 in the parenchyma might alter the sensitivity of cells to proliferative signals.

Finally, we observed that the SOX2+ stem cell compartment was expanded in the WT-TG but not PAT-TG postnatal and adult pituitary gland. This increase in stem cell number may be a result of 'sparing' of stem cell proliferation earlier in development, since WT-TG pituitaries have significantly fewer dividing cells in the marginal zone at P7. Regardless, the overall effect of increased DLK1 dosage is to increase overall AP volume and increase the size of the stem cell population.

In conclusion, we have shown that DLK1 is an important determinant of anterior pituitary size, and acts at multiple stages of development to modulate proliferative populations. The consequences of altered *Dlk1* dosage in the pituitary affect hormone production throughout life.

## Methods

### Mice

All animal procedures were carried out in accordance with the recommendations provided in the Animals (Scientific procedures) Act 1986 of the UK Government. Mice were maintained on a 12 hr light: dark cycle in a temperature and humidity-controlled room and re-housed in clean cages weekly. All mice were fed ad libitum and given fresh tap water daily. Mice were weaned at postnatal day (P) 21, or a few days later if particularly small. Thereafter, they were housed in single-sex groups (5 per cage maximum) or occasionally singly housed. Embryos were generated through timed matings (either as a pair or a trio). Noon on the day of the vaginal plug was considered as embryonic day (E)0.5.

All mice were maintained on a C57BL6/J background. The generation of the *Dlk1*-knockout (*Dlk1*^tm1Srba) and the *Dlk1*-TG BAC transgenic (Tg^*Dlk1-70C*) lines has been previously described (*Raghunandan et al., 2008*; *da Rocha et al., 2009*). For colony maintenance maternal heterozygotes (*Dlk1*^tm1Srbpa/+, hereafter called *Dlk1*^-/+ or MAT) and littermate wild-type (*Dlk1*^+/+, WT) animals were generated. Tg^*Dlk1-70C* (WT-TG) animals were maintained as heterozygotes by crossing WT females with heterozygous WT-TG males. *Dlk1*^+/tm1Srbpa (hereafter called *Dlk1*^+/- or PAT) and littermate WT animals or embryos were generated by crossing WT females with MAT males. *Dlk1*^+/tm1Srbpa; Tg^*Dlk1-70C* (PAT-TG) animals or embryos were obtained by crossing WT-TG females with MAT males. From these crosses, WT, PAT, and WT-TG littermates were also generated (shown in *Figure 2A*).

### Genotyping and *Dlk1* isoform analysis

Genotyping was performed on ear and embryonic tail biopsies with DNA extracted using DNAReleasy (LS02, Anachem). PCR was performed using REDTaq ReadyMix PCR Reaction Mix (R2523, Merck-SIGMA) using previously published primers (*Charalambous et al., 2014*).

### Hypothalamus isolation

Animals were culled by terminal injection of Pentobarbital (0.1 mL intraperitoneal EUTHATAL, Dopharma Research B. V. the Netherlands) followed by exanguination by cardiac puncture. The brain was removed and a 4 mm coronal midbrain section isolated using a 1 mm adult mouse brain matrix (Alto). A section containing the hypothalamus was isolated using the corpus callosum as the dorsal boundary, and the lateral ventricals as the lateral boundaries.

### RNA extraction and cDNA synthesis

RNA from pituitary glands of E18.5 embryos and adults, and adult hypothalamus was extracted using TRIzol LS (10296028, Invitrogen) and treated with DNase I (M0303, New England Biolabs [NEB]),

following the manufacturer's instructions. Complementary DNA (cDNA) was obtained by Reverse Transcription (RT) using 200–300 ng of purified RNA as template and Moloney Murine Leukemia Virus (M-MuLV) Reverse Transcriptase (M0253, NEB). cDNA was synthesised using the standard first strand synthesis protocol with random hexamers (S1230, NEB).

## Semi-quantitative PCR

*Dlk1* isoform usage was determined in embryonic pituitary cDNA as described (*Charalambous et al., 2014*), using *tuba* as a loading control.

## Quantitative real-time PCR

RT-qPCR was performed on cDNA isolated from adult hypothalamus using the Quantinova SYBR green kit (QIAGEN #208056) on a Quantstudio 6 thermocycler (Applied Biosystems). Data was analysed using the relative standard curve method as previously described (*Scagliotti et al., 2021*). *Actb* expression was used to normalise the expression of the target genes. Details of the primers used in this study are included in *Supplementary file 1i*.

## Histology

Fresh embryos and postnatal pituitaries were fixed with 4% w/v paraformaldehyde (PFA, P6148, Merck-SIGMA) in Phosphate-Buffered Saline (PBS, BR0014G, Thermo Scientific Oxoid) or Neutral Buffered Formalin (Merck-SIGMA, HT501128) overnight at 4 °C and dehydrated through an increasing ethanol series the following day. Samples were stored at 4 °C in 70% ethanol or dehydrated to 100% ethanol the day before the paraffin embedding. On the day of embedding, samples were incubated at room temperature (RT) with Histoclear II (National Diagnostics, HS202) (2x20 min for E9.5-E11.5 and postnatal pituitaries, 2x35 min for E13.5) or Xylene (VWR) (2x45 min for E15.5, 2x1 hr for E18.5). This was followed by 3x1 hr incubations at 65 °C with Histosec (1.15161.2504, VWR). Five µm histological sections were cut using a Thermo HM325 microtome, mounted on Menzel-Gläser Superfrost-Plus slides (Thermo Fisher Scientific, J1810AMNZ) and used for in situ hybridisation (ISH), RNAscope, immunohistochemistry (IHC) and immunofluorescence (IF).

## mRNA in situ hybridisation

ISH was performed as previously described (*Giri et al., 2017*). Sections were hybridised overnight at 65 °C with sense and antisense digoxigenin (DIG)-riboprobes against *Dlk1* (*da Rocha et al., 2009*). Sections were washed and incubated overnight at 4 °C with anti-Digoxigenin-AP antibody (45–11093274910 Merck-SIGMA, 1:1000). Staining was achieved by adding a solution of 4-Nitro blue tetrazolium chloride (NBT, 11383213001, Merck-SIGMA) and 5-Bromo-4-chloro-3-indolyl phosphate disodium salt (BCIP, 11383221001, Merck-SIGMA). Sections were mounted using DPX (6522, Merck-SIGMA). Sense controls for each probe were tested at E13.5 and showed no staining under identical conditions.

## RNAScope mRNA in situ hybridisation

RNAScope experiments were performed as previously described (*Russell et al., 2021*). For the expression analysis of early developmental markers, mRNA expression was assessed using the RNAScope singleplex chromogenic kits (Advanced Cell Diagnostics) on NBF fixed paraffin embedded sections processed as described in the previous section. The probes used for this experiment are listed in Supplemetary File 1 g. ImmEdge Hydrophobic Barrier PAP Pen (H-4000, Vector Laboratories) was used to draw a barrier around section while air-drying following the first ethanol washes. Sections were counterstained with Mayer's Haematoxylin (Vector Laboratories, H-3404), left to dry at 60 °C for 30 min before mounting with VectaMount Permanent Mounting Medium (Vector Laboratories, H-5000).

## Immunohistochemistry and immunofluorescence

IHC on histological sections was performed as previously described (*Giri et al., 2017*). For IHC, unmasking was achieved by boiling the histological sections with 10 mM tri-sodium citrate buffer pH 6 for 20 min. Detection of the proteins was achieved by incubating the histological sections overnight at 4 °C with the primary antibodies described in *Supplementary file 1h*, and detected with

secondary biotinylated goat α-rabbit, goat α-mouse or α -goat secondary (BA-1000, BA-9200 and BA9500, Vector Laboratories, 1:300), followed by 1 hr incubation at room temperature with Vectastain Elite ABC-HRP kit (PK-6100, Vector Laboratories). Staining was achieved through colorimetric reaction using DAB Peroxidase Substrate Kit (SK-4100, Vector Laboratories). Slides were lightly counterstained with Mayer's Haematoxylin (MHS16, Merck-SIGMA) and mounted using DPX. For double IF with hormonal markers an adult sections, unmasking was achieved as described above. For double IF on embryo sections, unmasking was performed with Tris-EDTA buffer pH 9 [10 mM Tris Base, 1 mM EDTA, 0.05% Tween 20]. Histological sections were incubated overnight at 4 °C with the primary antibody, incubated sequentially with a goat biotinylated anti-rabbit (1:200) then with streptavidin-594 (SA5594, Vector Laboratories, 1:200) at RT. The second primary was added to the sections overnight at 4 °C then detected with a goat α-rabbit DyLight 488 (DI1488, Vector Laboratories, 1:200) and mounted using VECTASHIELD Antifade Mounting Medium with DAPI (H-1200–10, Vector Laboratories). Isotype controls for each antibody showed no staining under identical conditions.

## Stereological estimation of pituitary volumes and cell number

Pituitary volume was estimated using the Cavallieri method (*Howard and Reed, 1998*). Briefly, the pituitary was exhaustively sectioned then sections were collected at regular, non-overlapping intervals throughout the gland from a random starting point and stained with H&E. Images of each H&E-stained section were acquired using a NanoZoomer HT (Hamamatsu) and processed using NDP.view2 software (Hamamatsu) to calculate the pituitary cross sectional area (CSA) at least 20 times per pituitary, which was then converted to volume. Samples were blinded prior to counting. Estimating total cell number: For each animal the maximal CSA was determined from the volumetric analysis above. At this level 3 independent sections were used to measure anterior pituitary CSA and count haemotoxylin-stained nuclei (using the cell detection tool in QuPath v0.3.2 [*Bankhead et al., 2017*], standard Hematoxylin OD settings except with threshold intensity set to 0.05), giving a mean cells/CSA measurement per individual. This number was multiplied by the AP volume to give the total cell number.

## Cell counting

Histological sections were immunostained as described above and imaged using the NanoZoomer HT (Hammamatsu). Images were exported using NDP.view2 and cell counted using the cell counter plugin in ImageJ (*Schindelin et al., 2012*). For IHC, cells that exhibited a clear brown staining were manually counted as positive. Samples were blinded prior to counting. Total number of cells was calculated by counting haematoxylin-counterstained cells. For IF, cells that exhibited a clear fluorescence signal were manually counted as positive. Total number of cells was calculated by counting DAPI-counterstained cells.

## Imaging

Images of histological sections were acquired using a NanoZoomer HT (Hammamatsu). Fluorescence images and higher magnification bright-field images were acquired using a Zeiss Axioplan II microscope with a Luminera 3 digital camera and INFINITY ANALYSE software v6.5.6 (Luminera). Images were combined using Adobe Photoshop 23.4.1 release 2022 (Adobe).

## Analyses of single cell and single nuclei RNA-seq data

Single-cell sequencing data from the mouse pituitary gland was downloaded from GEO using accession numbers GSE120410 (postnatal day 4, P4) and GSE142074 (postnatal day 49, P49) and analysed as previously described (*Scagliotti et al., 2021*).

Human single-nuclei RNA sequencing pituitary data was obtained from GEO accession number GSE178454. Using Seurat (v4.1.0) (*Satija et al., 2015*), (*Macosko et al., 2015*), (*Stuart et al., 2019*), (*Hao et al., 2021*) in R, cells from male and female datasets of all ages were taken forward if they expressed between 1000 and 5500 genes and <10% mitochondrial transcripts, removing doublets and low-quality cells. All 6 filtered datasets were integrated using the SCTransform workflow (*Hafemeister and Satija, 2019*). Clustering and visualisation for the integrated objects were carried out using the the default resolution and 1:15 principal components. Clusters were named according to known cell-type markers as previously reported (*Cheung et al., 2018*), (*Zhang et al., 2022*), (*Scagliotti et al., 2021*).

For both mouse and human data, *Dlk1/DLK1* expression was plotted in the respective datasets using the "FeaturePlot" function in Seurat with a min.cutoff=0 and split.by = 'Age'. Percentage of stem cells expressing *Dlk1/DLK1* was calculated under the 'integrated' assay of the Seurat object using the WhichCells function of grouped 'Stem Cells' expressing *Dlk1/DLK1*.

## Mapping POU1F1 binding sites to the mouse *Dlk1* region

POU1F1 binding sites identified using chromatin immunoprecipitation followed by sequencing (ChiP-seq) in growth hormone (GH)-expressing rat pituitary cell line were obtained from a published dataset (*Skowronska-Krawczyk et al., 2014*). POU1F1 binding sites overlapping the rat *Dlk1* locus (chr6: 133828590–134274323, rn4 assembly) were selected and converted to the orthologues region on mouse chr12 (mm10 assembly) using University of California Santa Cruz Genomics Institute Genome Browser (UCSC). The reported POU1F1 binding sites displayed a rat/mouse sequence conservation higher than 98%. Data were visualised using Gviz R package (*Hahne and Ivanek, 2016*).

## Western blotting

Sample preparation, gel electrophoresis and western blotting were described previously (*da Rocha et al., 2009*), with Abcam anti-DLK1 (ab21682, which recognises the intracellular domain of the protein) at 1:500. Anti-alpha tubulin (Merck-SIGMA, clone B-5-1-2) was used as a loading control at 1:10000.

## Pituitary SC cultures

Pituitary stem cells were isolated from anterior pituitaries collected from adult female mice at 13–16 weeks of age, essentially as described in *Andoniadou et al., 2013* (*Andoniadou et al., 2013*). Briefly, mice were culled by $CO_2$ asphyxiation, the skin overlying the skull cut and the skull bone opened with scissors. Using a dissecting microscope, the brain was carefully removed, as well as the posterior lobe of the pituitary gland. The remaining anterior pituitary was gently detached and transferred to a 1.5 ml microcentrifuge tube containing 200 µl of enzyme mix (0.5% w/v Collagenase type II [Worthington], 50 µg/ml DNase [Worthington], 2.5 µg/ml Fungizone [Gibco] and 0.1% v/v trypsin-EDTA solution 0.05% [Gibco] in Hank's Balanced Salt Solution [HBSS, Gibco]). Once transferred, the anterior pituitary was incubated at 37 °C for up to 4 hr, under gentle agitation. Following the incubation, samples were mechanically dissociated into single cell suspensions by vigorously pipetting the solution up and down. One ml of HBSS was added to the solution to dilute the enzyme mix and samples were centrifuged at 2500 rpm for 5 min. The cell pellet was re-suspended in Ultraculture Medium (Lonza), supplemented with 5% v/v fetal bovine serum (FBS, Invitrogen), 1% v/v penicillin/streptomycin (P/S, Merck-SIGMA), 20 ng/ml basic fibroblast growth factor (bFGF, R&D) and 50 ng/ml Cholera Toxin (Merck-SIGMA). Cells were counted using a haemocytometer, upon incubation with an equal volume of Trypan blue solution 0.4% (Merck-SIGMA) to assess cell viability. Viable cells were then plated at a density of 1000 cells/cm² and fresh media replaced every 3 days.

## Plasma GH determination

ELISA kits were used for measurements Growth Hormone (Merck-SIGMA EZRMGH-45K) according to manufacturers' instructions.

## Timing of puberty

Day of puberty was determined as the time of first vaginal opening (VO) in postnatal female mice. Animals were checked daily in the morning from P18 to P35 and the first appearance of VO and body weight on that day recorded. For this experiment all mice were weaned at P21.

## Statistical analysis

All statistical tests were performed using the GraphPad Prism Software version 9 for Windows, GraphPad Software, San Diego California USA, https://www.graphpad.com. Specific tests, significance values and number of samples analysed are indicated in the respective figure/table legends, and all error bars represent the standard deviation (SD). Data points in graphs represent individual animals as biological replicates. Outliers were removed using Grubb's test within the Prism application.

## Acknowledgements

We thank Prof Anne Ferguson-Smith, University of Cambridge for the $TG^{Dlk1-70C}$ mouse line. Funding was provided by the Medical Research Council (MRC) Grants MR/L002345/1 (MC), MR/R022836/1 (MC) and MR/T012153/1 (CLA), the Merck 2020 Grant for Growth Innovation (MC), and the Society for Endocrinology, UK (MH). MLV was supported by a studentship from the NIHR Biomedical Research Centre at Guy's and St Thomas' NHS Foundation Trust and King's College London. TLW was funded by King's College London as part of the "Cell Therapies and Regenerative Medicine" Four-Year Welcome Trust PhD Training Program. CGM was sponsored by Action Medical Research (GN2272) and Barts Charity (GN 417/2238 & MGU0551).

## Additional information

### Funding

| Funder | Grant reference number | Author |
|---|---|---|
| Medical Research Council | MR/L002345/1 | Mark Howard<br>Marika Charalambous |
| Medical Research Council | MR/R022836/1 | Valeria Scagliotti<br>Eugenia Marinelli<br>Marika Charalambous |
| Medical Research Council | MR/T012153/1 | Cynthia Andoniadou |
| Merck Healthcare KGaA | GGI 2020 | Valeria Scagliotti<br>Maria Lillina Vignola<br>Marika Charalambous |
| Society for Endocrinology | ECR Grant | Mark Howard |
| Guy's and St Thomas' NHS Foundation Trust | BRC-NIHR PhD studentship | Maria Lillina Vignola |
| King's College London | "Cell Therapies and Regenerative Medicine" Four-Year Welcome Trust PhD Training Program | Thea Willis |
| Action Medical Research | GN2272 | Carles Gaston-Massuet |
| Barts Charity | GN 417/2238 | Carles Gaston-Massuet |
| Barts Charity | MGU0551 | Carles Gaston-Massuet |

The funders had no role in study design, data collection and interpretation, or the decision to submit the work for publication.

### Author contributions

Valeria Scagliotti, Data curation, Formal analysis, Investigation, Visualization, Methodology, Writing – original draft; Maria Lillina Vignola, Formal analysis, Investigation, Visualization, Methodology, Writing – original draft; Thea Willis, Formal analysis, Investigation, Methodology, Writing – original draft; Mark Howard, Formal analysis, Investigation, Visualization, Methodology, Writing – review and editing; Eugenia Marinelli, Investigation, Writing – review and editing; Carles Gaston-Massuet, Resources, Investigation, Methodology, Writing – review and editing; Cynthia Andoniadou, Conceptualization, Resources, Formal analysis, Supervision, Funding acquisition, Writing – original draft, Project administration; Marika Charalambous, Conceptualization, Resources, Data curation, Formal analysis, Supervision, Funding acquisition, Investigation, Visualization, Methodology, Writing – original draft, Project administration

### Author ORCIDs

Maria Lillina Vignola http://orcid.org/0000-0001-7121-7715
Thea Willis https://orcid.org/0000-0002-1794-7490
Cynthia Andoniadou https://orcid.org/0000-0003-4311-5855

Marika Charalambous http://orcid.org/0000-0002-1684-5783

**Decision letter and Author response**
Decision letter https://doi.org/10.7554/eLife.84092.sa1
Author response https://doi.org/10.7554/eLife.84092.sa2

## Additional files

### Supplementary files

• Supplementary file 1. Quantitative assessment of pituitary size and hormone cell composition in Dlk1-overexpressing adult mice. (**a**) Proportions of AP cells labelled with hormonal markers in WT and WT-TG female animals at 12 weeks of age. (**b**) Total body mass of animals from matched litters sacrificed from E11.5 to P21. Individuals in each age group were compared by One-Way ANOVA with post-hoc pairwise testing WT vs WT-TG, PAT, PAT-TG and PAT vs PAT-TG, corrected for multiple comparisons using Bonferroni's adjustment. (**c**) Pituitary volumes acquired by stereological estimation in the embryo. Data from males and females is combined. (**d**) Pituitary volumes acquired by stereological estimation in of the intact postnatal gland. Individuals in each age group were compared by One-Way ANOVA with post-hoc pairwise testing WT vs WT-TG, PAT, PAT-TG and PAT vs PAT-TG, corrected for multiple comparisons using Bonferroni's adjustment. Data from males and females is combined. (**e**) Pituitary volumes acquired by stereological estimation in of the intact adult gland. All animals were compared by Two-Way ANOVA with post-hoc pairwise testing WT vs WT-TG, PAT, PAT-TG and PAT vs PAT-TG, using Dunnett's multiple comparison test. (**f**) Proportion of proliferating cells (IHC positive for p-HH3) in the embryonic and postnatal pituitary gland. Data from males and females is combined. (**g**) RNAScope probes used in this study. (**h**) Primary antibodies used in the study. (**i**) RT-PCR and RT-qPCR primers used in this study.

• MDAR checklist

• Source data 1. Zipped file containing raw data used to generate graphs in *Figures 5 and 6*. Fig5ABC&FigAB_pHH3. Cell counts of pHH3+ cells in developmental time course. Fig5G_POU1F1. Cell counts of POU1F1+ cells at e13.5. Fig5H_HES1. Cell counts of HES1+ cells at e13.5. Fig6DE_SOX2. Cell counts for %SOX2+ cells and SOX2+ compartmental volume at postnatal day 21.

### Data availability

Sequencing data have previously been deposited in GEO under accession codes GSE120410, GSE142074, GSE178454. Figure 1 - source data 1, Figure 4 - source data 1 and Source data 1 contain the numerical data used to generate the figures.

The following previously published datasets were used:

| Author(s) | Year | Dataset title | Dataset URL | Database and Identifier |
|---|---|---|---|---|
| Cheung LYM, George AS, McGee SR, Daly AZ | 2018 | Single-cell RNA sequencing reveals novel markers of pituitary stem cells and hormone-producing cell-types | https://www.ncbi.nlm.nih.gov/geo/query/acc.cgi?acc=GSE120410 | NCBI Gene Expression Omnibus, GSE120410 |
| Cheung LY, Camper SA | 2020 | PROP1-Dependent Retinoic Acid Signaling Regulates Developmental Pituitary Morphogenesis and Hormone Expression | https://www.ncbi.nlm.nih.gov/geo/query/acc.cgi?acc=GSE142074 | NCBI Gene Expression Omnibus, GSE142074 |
| Zamojski M | 2022 | Single nucleus pituitary transcriptomic and epigenetic landscape reveals human stem cell heterogeneity with diverse regulatory mechanisms | https://www.ncbi.nlm.nih.gov/geo/query/acc.cgi?acc=GSE178454 | NCBI Gene Expression Omnibus, GSE178454 |

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

# Appendix 1

**Appendix 1—key resources table**

| Reagent type (species) or resource | Designation | Source or reference | Identifiers | Additional information |
|---|---|---|---|---|
| Gene (*Mus musculus*) | Delta-like homologue 1 (Dlk1) | MGI:94900 | Dlk1 | |
| Strain, strain background (*Mus musculus*, males and female) | C57BL6J | | WT | |
| Genetic reagent (*Mus musculus*) | Dlk1tm1Srba | **Raghunandan et al., 2008** MGI:3526402 | "Dlk1 deletion; PAT; PAT-TG" | |
| Genetic reagent (*Mus musculus*) | TgDlk1-70 | **da Rocha et al., 2009** | "Dlk1 transgenic; WT-TG; PAT-TG" | |
| Antibody | Anti-SOX2 (Goat Polyclonal) | Immune Systems Ltd | Anti-SOX2 | GT15098, RRID:AB_2195800 |
| Antibody | Anti-SOX2(Rabbit Monoclonal) | Abcam | Anti-SOX2 | ab92494, RRID:AB_10585428 |
| Antibody | Anti-POU1F1 (PIT1) (Rabbit Monoclonal) | Gifted by Dr S. J. Rhodes (IUPUI, USA) | Anti-POU1F1 | 422_Rhodes, RRID:AB_2722652 |
| Antibody | Anti-pHH3 (Rabbit Polyclonal) | Millipore | Anti-pHH3 | 05–806, Anti-phospho-Histone H3 (Ser10) Antibody, clone 3 H10 https://www.merckmillipore.com/GB/en/product/Anti-phospho-Histone-H3-Ser10-Antibody-clone-3H10,MM_NF-05–806 |
| Antibody | Anti-GH (Rabbit Polyclonal) | National Hormone and Peptide Program (NHPP) | Anti-GH | AFP-5641801 |
| Antibody | Anti-TSH (Rabbit Polyclonal) | National Hormone and Peptide Program (NHPP) | Anti-TSH | AFP-1274789 |
| Antibody | Anti-PRL (Rabbit Polyclonal) | National Hormone and Peptide Program (NHPP) | Anti-PRL | AFP-4251091 |
| Antibody | Anti-ACTH (Mouse Monoclonal) | National Hormone and Peptide Program (NHPP) | Anti-ACTH | AFP-156102789 |
| Antibody | Anti-DLK1 (Rabbit monoclonal) | abcam | Anti-DLK1 | ab21682 https://www.abcam.com/products/primary-antibodies/dlk-1-antibody-ab21682.html |
| Antibody | Anti-DLK1 (Goat polyclonal) | R&D | AF8277 | AF8277 https://www.rndsystems.com/products/mouse-pref-1-dlk1-fa1-antibody_af8277 |
| Antibody | Anti-HES1 | Cell Signaling Technologies | Anti-HES1 | D6P2U |
| Antibody | Anti-Rabbit 488 (Goat Polyclonal) | Life Technologies | Anti-rabbit 488 | A11008, RRID:AB_143165 |
| Antibody | Anti-Rabbit 647 (Goat Polyclonal) | Life Technologies | Anti-rabbit 594 | A21050, RRID:AB_141431 |
| Antibody | anti-Digoxigenin-AP | Millipore-SIGMA | Anti-DIG | 45–11093274910 |
| Antibody | Anti-alpha tubulin | Millipore-SIGMA | Anti-tub | T5168 |
| Antibody | Anti-Goat 488 (Donkey Polyclonal) | Abcam | Anti-goat 488 | ab150133, RRID:AB_2832252 |
| Antibody | biotinylated goat α-rabbit | Vector Laboratories | biotinylated goat α-rabbit | BA-1000 |

*Appendix 1 Continued on next page*

*Appendix 1 Continued*

| Reagent type (species) or resource | Designation | Source or reference | Identifiers | Additional information |
|---|---|---|---|---|
| Antibody | biotinylated goat α-mouse | Vector Laboratories | biotinylated goat α-mouse | BA-9200 |
| Antibody | biotinylated α -goat | Vector Laboratories | biotinylated α -goat | BA9500 |
| Sequence-based reagent | RNAscope probe *M. musculus* Axin2 | Advanced Cell Diagnostics | Mm-Axin2 | 400331 |
| Sequence-based reagent | RNAscope probe *M. musculus* Shh | Advanced Cell Diagnostics | Mm-Shh | 314361 |
| Sequence-based reagent | RNAscope probe *M. musculus* Fgf8 | Advanced Cell Diagnostics | Mm-Fgf8 | 313411 |
| Sequence-based reagent | RNAscope probe *M. musculus* Fgf10 | Advanced Cell Diagnostics | Mm-Fgf10 | 446371 |
| Sequence-based reagent | RNAscope probe Lef1 | Advanced Cell Diagnostics | Mm-Lef1 | 441861 |
| Sequence-based reagent | Dlk1 qPCR Forward primer | PMID:25349437 | | GAAAGGACTGCCAGCACAAG |
| Sequence-based reagent | Dlk1 qPCR Reverse primer | PMID:25349437 | | CACAGAAGTTGCCTGAGAAGC |
| Sequence-based reagent | Dlk1 splice qPCR Forward primer | PMID:25349437 | | CTGCACACCTGGGTTCTCTG |
| Sequence-based reagent | Dlk1 splice qPCR Reverse primer | PMID:25349437 | | CTGCACACCTGGGTTCTCTG |
| Sequence-based reagent | Ghrh qPCR Forward primer | This paper | | GCTGTATGCCCGGAAAAGTGAT |
| Sequence-based reagent | Ghrh qPCR Reverse primer | This paper | | AATCCCTGCAAGATGCTCTCC |
| Sequence-based reagent | Sst qPCR Forward primer | This paper | | CCCAGACTCCGTCAGTTTCT |
| Sequence-based reagent | Sst qPCR Reverse primer | This paper | | GGGCATCATTCTCTGTCTGG |
| Sequence-based reagent | Actb qPCR Forward primer | PMID:25349437 | | TTCTTTGCAGCTCCTTCGTT |
| Sequence-based reagent | Actb qPCR Reverse primer | PMID:25349437 | | ATGGAGGGGAATACAGCCC |
| Sequence-based reagent | Tuba qPCR Forward primer | PMID:25349437 | | AGACCATTGGGGGAGGAGAT |
| Sequence-based reagent | Tuba qPCR Reverse primer | PMID:25349437 | | GTGGGTTCCAGGTCTACGAA |
| Commercial assay or kit | RNAScope 2.5 HD Assay-RED | Advanced Cell Diagnostics | | 322350 |
| Commercial assay or kit | ABC kit | Vector Laboratories | | Cat# Vector PK-6100 RRID:AB_2336819 |
| Commercial assay or kit | BCA assay | Thermo Fisher | Cat# 23227 | |
| Commercial assay or kit | REDTaq ReadyMix PCR Reaction Mix | Sigma-Aldrich (Merck) | R2523 | |
| Software, algorithm | Prism 9 | GraphPad Software | | https://www.graphpad.com/ |
| Software, algorithm | NDP View | Hamamatsu Photonics | | https://www.hamamatsu.com/ |

*Appendix 1 Continued on next page*

*Appendix 1 Continued*

| Reagent type (species) or resource | Designation | Source or reference | Identifiers | Additional information |
| --- | --- | --- | --- | --- |
| Software, algorithm | The Galaxy Platform | *Afgan et al., 2016*; *Blankenberg et al., 2010*; *Goecks et al., 2010* | | https://usegalaxu.org RRID:SCR_006281 |
| Software, algorithm | DESeq2 v2.11.38 | *Love et al., 2014* | | https://github.com/Bioconductor-mirror/DESeq2 RRID:SCR_015687 |
| Software, algorithm | featureCounts v1.4.6p5 | *Liao et al., 2014* | | http://subread.sourceforge.net/ RRID:SCR_012919 |
| Software, algorithm | QuPath | PMID:29203879 | | https://qupath.readthedocs.io/en/0.4/ |

