## [Editor Report]

This fundamental work substantially advances our understanding of the role of Dlk1 in pituitary gland size and implicating WNT pathway. The evidence supporting the conclusions is compelling, with rigorous mouse genetic models and state-of-the-art ChipSeq and scRNA seq methods. The work will be of broad interest to cell biologists, developmental biologists and neuroendocrinologists.

---

## [Decision Letter]

**Decision letter after peer review:**

Thank you for submitting your article "Imprinted Dlk1 dosage as a size determinant of the mammalian pituitary gland" for consideration by *eLife*. Your article has been reviewed by 2 peer reviewers, and the evaluation has been overseen by a Reviewing Editor and Edward Morrisey as the Senior Editor. The following individual involved in the review of your submission has agreed to reveal their identity: Leonard Cheung (Reviewer #1).

Essential revisions:

1) Clarification of hyperplasia or hypertrophy contributing to enlarged pituitary size. Cell number representations, particularly in GH, PRL and TSH populations.

2) Data presented in Figure 3 needs to be explained with regard to Dlk1 expression in the MZ compartment.

3) Pituitary size in PatTg mice at later time points in adulthood needs to be determined.

4) Details about the contribution of Dlk1 in the diencephalon/hypothalamus to pituitary organ size.

5) Role of Wnt signaling in Pat and which pathways are involved.

6) Clarity about GH action in postnatal growth.

7) Sex differences are not considered for adult time points. The authors may include data from males.

8) Include missing Details on methods and other aspects directly in the main text.

9) Include details on PRL immunostaining protocol-antigen retrieval conditions.

10) Timing of Pou1f1 and Gh1 expression needs to be described.

*Reviewer #1 (Recommendations for the authors):*

I had a question about the expression of Dlk1 in the MZ *sox2*^+^ compartment as shown in Figure 3A. The expression in the WT (inset) shows strong staining along the marginal zone/basal membrane, and this is clearly not detected in the Pat and Pat-Tg. At the same time, I noticed that Figure 3E and 2E may not show the same (basal) pattern, although the magnifications are different. In Cheung 2013 JNE there appeared to be little Dlk1 immunostaining in cells along the MZ, although that was a single panel. Could you elaborate on the expression pattern of Dlk1 in the MZ compartment? I agree that Figure 3E (WT inset) shows intercellular membranous Dlk1 staining in contrast to the cytoplasmic staining seen in Pou1f1+ cells and that this appears to be missing in the Pat-Tg inset.

I believe some caution may be considered in using single-cell data in Figure 3C/D. Since Dlk1 is expressed in many Gh cells and somatotropes commonly lyse and contaminate other cell types in sc data, at least part of that Dlk1 detected in stem cells may be caused by cell-free RNA. In Figure 3C, you can see some Dlk1 detected in the endothelia (presumably EC) and connective tissue (presumed CT) in the P49 data. At the same time, I recognize that the P4 data do not show Dlk1 in the corresponding cells, suggesting contamination may be limited.

In the context of Figure S1, you could reference Ansell 2007 Mol Cell Endo, which studied Dlk1 regulation of the GH promoter and expression, finding that Dlk1 in vitro represses Gh expression and its promoter and that it also represses Pou1f1 action when cotransfected.

I had a question about the reduced pituitary size of Pat and Pat-Tg animals. While they are clearly reduced before P21, both are recovering (non-sig) by P21. Do you know if the pituitary continues to be smaller in later adulthood? If not, could the reduced size in early life be a developmental delay rather than a permanent reduction in organ size? Puertas-Avendano 2011 studied 4-month-old mice and did not study pituitary size (but did not report it). Cheung 2013 used P14-P100 mice and looked at total pituitary protein at P42 as a measure of the pituitary size and did not find a difference (size was also not measured).

I was wondering about the contribution of Dlk1 in the diencephalon/hypothalamus to pituitary organ size in your models. You and others (Allen Brain Atlas, Villanueva 2011 PLOS One, Persson-Augner 2014 Neuroendo) show strong expression of Dlk1 in the diencephalon and arcuate nucleus. While you show in Figure S3 that signals in the developing ventral diencephalon are normal, I am unfamiliar with the data regarding the effects of Dlk1 loss/overexpression on these cells postnatally. GHRH-overexpressing transgenic mice (Stefaneanu 1989 Endo) have GH and PRL cell hypertrophy and hyperplasia and did not appear to cause adenoma by 8 months, which are partially similar phenotypes to your WT-Tg. Do you know if GHRH or other pituitary-regulating hormone neurons are normal postnatally in the WT-Tg? I wonder if increased GHRH action could in part cause the increased serum GH found in Wt-Tg in Charalambous 2014 and be partially causative of increased pituitary size in WT-Tg in this study.

In Figure 5E/G, you observe that there are more Pou1f1+ cells at e13.5 in the Pat. Do you know if these persist postnatally? Puertas-Avendano 2011 and Cheung 2013 generally reported decreases in pituitary hormones in postnatal Dlk1-null.

In Figure 5I you show that Wnt signals are increased in Pat, potentially driving early differentiation and increase of Pou1f1+ cells, while the pituitary is markedly smaller in volume at e13.5 (Figure 4C). In a model with mildly increased Wnt signaling, Hesx1-Cre; Apc f/f (Gonzalez-Meljem 2017), the pituitaries show mild hyperplasia and some SASP/NFKB/DDR marker expression. Do you think any of these pathways may be activated in the Pat as a result of increased Wnt signals? Why might increased Wnt cause different size phenotypes in the two models?

On line 448, please verify whether later postnatal GH action can induce growth. I recall anecdotally that it can because bone epiphysial fusion occurs later in mice than humans but I admit I cannot find a reference for it. Mice treated with GH at 2 weeks of age appear to grow (Panici 2010 Faseb J 24(12)).

On lines 448-450 about GH pulsatility, please verify whether pulsatility is required for endogenous GH only. I thought exogenous hGH injections (daily) or bGH transgenic mice (constant) can induce growth without pulsatility.

*Reviewer #2 (Recommendations for the authors):*

1. Some information was not readily available in the manuscript and required reading the source data. Including this information in the manuscript itself will improve its readability:

a. For embryonic and postnatal time points please explain in the manuscript that sexes were pooled.

b. In figure 4C the y-axis is "Proportional mean of WT volume". In figure 4A the y-axis is labeled "Proportion of mean WT weight". Please explain more clearly in the manuscript that these measurements are proportional to WT.

2. In figure 1D, the PRL+ cells appear oddly small in the WT. PRL IHC looks more normal in the WT-TG. Is this a representative stain? Is it possible you are detecting something other than PRL? Were these samples also boiled in Tris-EDTA buffer for 20 minutes? This antigen retrieval method may be too severe for PRL.

3. The authors should define PSC.

4. Line 342 – the authors state that pituitary cells migrate "dorsally" from the lumen after exiting the cell cycle. I believe they migrate "ventrally".

5. Line 345 – should be Figure 3D, not Figure 2D.

6. Figure 4B: PAT and PAT-TG pituitary glands don't look 40-50% smaller than WT and WT-TG at e13.5. Are these representative images? Is the loss in volume more apparent rostrally and/or caudally?

7. Loss of Dlk1 leads to an increase in differentiating cells at the expense of stem cell maintenance. It would be interesting to know if the timing of the Pou1f1 or Gh1 expression changes.

8. Figure 6E – the *sox2*^+^ MZ volume was calculated using the Cavalieri method. How was the MZ identified in order to distinguish it from the parenchyma?

9. No description of the methods for RTqPCR is present.

---

## [Author Response]

Essential revisions:1) Clarification of hyperplasia or hypertrophy contributing to enlarged pituitary size. Cell number representations, particularly in GH, PRL and TSH populations.

We believe that the enlarged gland in WT-TG mice occurs through hyperplasia. This point has now been clarified throughout and new data added in Figure 4E. Here we used stereological estimation of total cell number in the adult anterior pituitary of all 4 study genotypes (WT, WT-TG, PAT, PAT-TG). WT-TG animals had increased AP cell number compared to WT littermates (p = 0.0443), and both PAT and PAT-TG animals had reduced cell number (p = 0.0212, 0.0193 respectively).

Regarding the apparent discrepancy in hormone cell proportions of cells highlighted by reviewer 1, we apologise for the confusion. We believe that the volume occupied by all hormone producing cells is increased in the WT-TG animals, consistent with the maintenance of cells proportions shown in Figure 1D. We think that the reason that only the GH population shows a volume expansion in Figure 1G is a problem of power. Please see our comment to reviewer 1 below. However, we have now performed 2-Way ANOVA on the data in Figure 1G, and this test clearly indicated that hormone cell volumes between groups significantly differ by genotype (p = 0.0009). The legend to Figure 1 has been modified to reflect this.

2) Data presented in Figure 3 needs to be explained with regard to Dlk1 expression in the MZ compartment.

We agree that there appears to be some heterogeneity in the postnatal MZ staining for DLK1. Looking closely at the P21 WT image in Figure 2E it is clear that there is more DLK1 in the medial MZ compared to the lateral part. The images in Figure 3B are all in the lateral part of the MZ. This may reflect important differences between stem cell populations that are currently a focus of the Andoniadou lab, but are beyond the scope of this paper.

3) Pituitary size in PatTg mice at later time points in adulthood needs to be determined.

We have now bred a new cohort of male and female animals from all 4 genotypes and sacrificed at 8 weeks to address the long-term influence of *Dlk1* ablation and the ability of the *Dlk1-70C* transgene to rescue the pituitary volume phenotype. These new data are shown in Figure 4D and E. Consistent with the data shown in Figure 4C, adult WT-TG animals of both sexes have increased AP volume and cell number. PAT males have a significant reduction in AP volume, but the female gland has reached WT volume. The volume of the AP in PAT-TG in both sexes is similar to PAT, confirming that both stem cell- and parenchymal-derived *Dlk1* is required for postnatal hyperplasia.

4) Details about the contribution of Dlk1 in the diencephalon/hypothalamus to pituitary organ size.

We tested levels of *Dlk1*, *Ghrh* and *Sst* in the hypothalamus of all 4 genotypes of adult mice. *Dlk1* is not expressed from the transgene in this tissue (also seen in Figure 2D) therefore a direct action for *Dlk1* in the hypothalamus of WT-TG mice to mediate pituitary volume expansion is less likely. Concurrently, expression of *Ghrh* and *Sst* was not different from WT in any genotype. See below comments to reviewer 1 for more details. These data have now been added to Figure 4F.

5) Role of Wnt signaling in Pat and which pathways are involved.

We find the remark of Reviewer 1 extremely interesting and have discussed the possible involvement of alternate pathways in the specific comment.

6) Clarity about GH action in postnatal growth.

Regarding the action of GH on postnatal growth dynamics, we have taken much of our information from the pioneering work of the Efstradiadis lab. They carefully measured growth rates in mice that were lacking components of the GH-IGF system, including the GH receptor (Ghr). In 2001 Lupu et al. showed that the majority of the body mass difference between WT and Ghr-/- mice occurs due to a reduction in growth rate prior to P21 (about 80% of the overall effect). By P30 Ghr-/- mice reach ~50% WT mass and remain at this relative mass until at least P110 (PMID: 11133160). From this we conclude that the normal period of growth promoting action by GH in mice is in the first 4 postnatal weeks. We stated in the discussion “while we did observe increased circulating GH in WT-TG animals at P21 (Figure S2), this may have been too late to promote the pre-pubertal growth spurt.” We stand by this statement. For clarity we have added the following line to the discussion “In mice, the majority of GH action on linear growth occurs prior to P21 (Lupu et al. 2001)”.

However, it must be noted that elevated GH administration to rodents can increase postnatal growth rates. This topic, with its caveats (human GH can act on PRLR, for example) is ably covered by Kopchick et. al (PMID: 24035867) in a 2014 review. From this it is clear that GH dosage has a major impact on its ability to stimulate growth. Indeed an early paper from Palmiter et. al (PMID: 6958982), where a rat GH transgene was introduced into mice by pronuclear injection, developed an allelic series where increased transgene copy number resulted in higher GH circulating levels and impact on growth. Importantly, a robust effect on growth was only observed when GH levels exceeded 100x normal circulating levels. From this we conclude (as have others, including Huang et. al 2019 -cited in discussion) that it is the pattern of GH secretion at near-physiological levels, rather than the dosage alone, that is crucial for the growth promoting effects of this hormone. This requirement can be overcome by extremely high GH dosage.

7) Sex differences are not considered for adult time points. The authors may include data from males.

We have included adult data for both sexes in Figure 4D. The minimal sexual dimorphism in *Dlk1*-mediated phenotypes match our earlier studies where altered body composition, reduced hepatic lipid deposition and elevated GH was observed in both sexes of WT-TG mice (PMID: 25349437).

8) Include missing Details on methods and other aspects directly in the main text.

We have done this – see comments in reply to reviewer 2.

9) Include details on PRL immunostaining protocol-antigen retrieval conditions.

We have done this – see comments in reply to reviewer 2.

10) Timing of Pou1f1 and Gh1 expression needs to be described.

We have done this – GH-positive somatotrophs are observed earlier in development in PAT compared to WT pituitaries, consistent with our hypothesis. Data has been added to Figure 5J. Please see additional comments to reviewer 2.

Reviewer #1 (Recommendations for the authors):I had a question about the expression of Dlk1 in the MZ sox2^+^ compartment as shown in Figure 3A. The expression in the WT (inset) shows strong staining along the marginal zone/basal membrane, and this is clearly not detected in the Pat and Pat-Tg. At the same time, I noticed that Figure 3E and 2E may not show the same (basal) pattern, although the magnifications are different. In Cheung 2013 JNE there appeared to be little Dlk1 immunostaining in cells along the MZ, although that was a single panel. Could you elaborate on the expression pattern of Dlk1 in the MZ compartment? I agree that Figure 3E (WT inset) shows intercellular membranous Dlk1 staining in contrast to the cytoplasmic staining seen in Pou1f1+ cells and that this appears to be missing in the Pat-Tg inset.

We agree that there appears to be some heterogeneity in the postnatal MZ staining for DLK1. See point 2 above.

I believe some caution may be considered in using single-cell data in Figure 3C/D. Since Dlk1 is expressed in many Gh cells and somatotropes commonly lyse and contaminate other cell types in sc data, at least part of that Dlk1 detected in stem cells may be caused by cell-free RNA. In Figure 3C, you can see some Dlk1 detected in the endothelia (presumably EC) and connective tissue (presumed CT) in the P49 data. At the same time, I recognize that the P4 data do not show Dlk1 in the corresponding cells, suggesting contamination may be limited.

We agree that any single data source has limitations. However, in this case the sc-RNAseq data is supported by IHC experiments.

In the context of Figure S1, you could reference Ansell 2007 Mol Cell Endo, which studied Dlk1 regulation of the GH promoter and expression, finding that Dlk1 in vitro represses Gh expression and its promoter and that it also represses Pou1f1 action when cotransfected.

We have added the following sentence to the discussion: *Dlk1* may be part of an autoregulatory feedback loop since, at least in vitro, expression of *Dlk1* represses the action of POU1F1 on the *Gh* promoter (Ansell 2007).

I had a question about the reduced pituitary size of Pat and Pat-Tg animals. While they are clearly reduced before P21, both are recovering (non-sig) by P21. Do you know if the pituitary continues to be smaller in later adulthood? If not, could the reduced size in early life be a developmental delay rather than a permanent reduction in organ size? Puertas-Avendano 2011 studied 4-month-old mice and did not study pituitary size (but did not report it). Cheung 2013 used P14-P100 mice and looked at total pituitary protein at P42 as a measure of the pituitary size and did not find a difference (size was also not measured).

We have now included these data, see point 3 above.

I was wondering about the contribution of Dlk1 in the diencephalon/hypothalamus to pituitary organ size in your models. You and others (Allen Brain Atlas, Villanueva 2011 PLOS One, Persson-Augner 2014 Neuroendo) show strong expression of Dlk1 in the diencephalon and arcuate nucleus. While you show in Figure S3 that signals in the developing ventral diencephalon are normal, I am unfamiliar with the data regarding the effects of Dlk1 loss/overexpression on these cells postnatally. GHRH-overexpressing transgenic mice (Stefaneanu 1989 Endo) have GH and PRL cell hypertrophy and hyperplasia and did not appear to cause adenoma by 8 months, which are partially similar phenotypes to your WT-Tg. Do you know if GHRH or other pituitary-regulating hormone neurons are normal postnatally in the WT-Tg? I wonder if increased GHRH action could in part cause the increased serum GH found in Wt-Tg in Charalambous 2014 and be partially causative of increased pituitary size in WT-Tg in this study.

We agree that alterations for hypothalamic production of GH-modifying neuropeptides such as GHRH and SST could potentially explain the GH overproduction in WT-TG mice, especially since DLK1 is normally expressed in the hypothalamic neurons that produce these signals. However, we had previously disregarded this possibility since we know that the *Dlk1-70C* transgene lacks the enhancer sequences necessary for hypothalamic gene expression (see the PAT-TG in Figure 2D for example). To more robustly test this hypothesis we performed RT-qPCR on hypothalamus samples from our new adult cohort. As expected, *Dlk1* expression was not modified by the transgene – WT == WT-TG and PAT == PAT-TG. Moreover, *Ghrh* and *Sst* gene expression was not significantly different between genotypes. We conclude that WT-TG animals do not appear to have a defect in GH-modifying neurohormone production, at least at the transcriptional level (which would also likely reflect a change in the size of the neuronal population). These data have been added as Figure 4F.

In Figure 5E/G, you observe that there are more Pou1f1+ cells at e13.5 in the Pat. Do you know if these persist postnatally? Puertas-Avendano 2011 and Cheung 2013 generally reported decreases in pituitary hormones in postnatal Dlk1-null.

We have not determined this quantitatively. However, POU1F1 expression in PAT and PAT-TG mice at e18.5 appears normal (Figure 3E, Figure S2M-P), suggesting to us that the critical period is around e13.5, and further changes to gland development in *Dlk1*-ablated pituitaries are consequential.

In Figure 5I you show that Wnt signals are increased in Pat, potentially driving early differentiation and increase of Pou1f1+ cells, while the pituitary is markedly smaller in volume at e13.5 (Figure 4C). In a model with mildly increased Wnt signaling, Hesx1-Cre; Apc f/f (Gonzalez-Meljem 2017), the pituitaries show mild hyperplasia and some SASP/NFKB/DDR marker expression. Do you think any of these pathways may be activated in the Pat as a result of increased Wnt signals? Why might increased Wnt cause different size phenotypes in the two models?

This is a very interesting observation, and we believe that the spatial, temporal and dosage parameters for WNT activation may lead to varied phenotypes. The WNT activation presented here in the Pat, is not as pronounced as in the Apc deletion model. The mild hyperplasia in the Apc deletion model is largely due to cell-autonomous activation of WNT signalling (in the absence of WNT ligand), which leads to over-proliferation of embryonic progenitors at early stages (from RP specification). Additional pathway elevation in the Apc deletion model is linked specifically to the SASP (oncogene-induced SASP), for which we have no evidence that it is taking place in the Pat. In the Pat we see reduced proliferation, which is not consistent with a SASP phenotype and associated pathway elevation, which would induce proliferation in non-senescent cells. In other WNT-activation models (e.g. Ctnnb1lox(ex3)/+), pituitary stem/progenitor cells enter SASP and express pathogenic levels of pathways (higher than in the Apc deletion), leading to cell non-autonomous tumours due to these SASP-related activities. On the other hand, WNT signalling that is either very high or very low, can both lead to defects in cell lineage commitment e.g. Ctnnb1-lox(ex3) expression paradoxically leads to complete absence of Pou1f1-lineage commitment (Andoniadou et al. 2013), just like a reduction in WNT signalling. In future, we would like to carry out (i) transcriptomic analyses in the Pat and other genotypes, and (ii) lineage-specific Dlk1 deletions, which together would help us resolve the downstream mechanisms and delineate additional pathway involvement.

On line 448, please verify whether later postnatal GH action can induce growth. I recall anecdotally that it can because bone epiphysial fusion occurs later in mice than humans but I admit I cannot find a reference for it. Mice treated with GH at 2 weeks of age appear to grow (Panici 2010 Faseb J 24(12)).On lines 448-450 about GH pulsatility, please verify whether pulsatility is required for endogenous GH only. I thought exogenous hGH injections (daily) or bGH transgenic mice (constant) can induce growth without pulsatility.

Please see our reply to point 6, above.

Reviewer #2 (Recommendations for the authors):1. Some information was not readily available in the manuscript and required reading the source data. Including this information in the manuscript itself will improve its readability:a. For embryonic and postnatal time points please explain in the manuscript that sexes were pooled.

Added to legend of Tables S3A, B and S4: Data from males and females is combined.

In the Figure 4 legend “sexes are combined since body weight is not sexually dimorphic at these stages”.

b. In figure 4C the y-axis is "Proportional mean of WT volume". In figure 4A the y-axis is labeled "Proportion of mean WT weight". Please explain more clearly in the manuscript that these measurements are proportional to WT.

This has been added and the figure legend clarified.

2. In figure 1D, the PRL+ cells appear oddly small in the WT. PRL IHC looks more normal in the WT-TG. Is this a representative stain? Is it possible you are detecting something other than PRL? Were these samples also boiled in Tris-EDTA buffer for 20 minutes? This antigen retrieval method may be too severe for PRL.

The WT PRL IHC image has been replaced with a more representative image from the same experiment. In fact, the antigen retrieval was performed with Citrate buffer for the DLK1-hormone experiments. We apologise for this mistake. This has been amended in the methods section “For double IF with hormonal markers in adult sections, unmasking was achieved as described above. For double IF on embryo sections, unmasking was performed with Tris-EDTA buffer pH 9 [10 mM Tris Base, 1 mM EDTA, 0.05% Tween 20].”

3. The authors should define PSC.

This was defined in line 38.

4. Line 342 – the authors state that pituitary cells migrate "dorsally" from the lumen after exiting the cell cycle. I believe they migrate "ventrally".

Thanks, this has been changed.

5. Line 345 – should be Figure 3D, not Figure 2D.

Changed.

6. Figure 4B: PAT and PAT-TG pituitary glands don't look 40-50% smaller than WT and WT-TG at e13.5. Are these representative images? Is the loss in volume more apparent rostrally and/or caudally?

Yes these are representative images. The majority of the volume loss appears to be in the medio-lateral axis, as can be seen at later stages in the same figure.

7. Loss of Dlk1 leads to an increase in differentiating cells at the expense of stem cell maintenance. It would be interesting to know if the timing of the Pou1f1 or Gh1 expression changes.

We could not observe POU1F1 earlier than e13.5 (we checked at e12.5), though a more precise timing study may be required to fully test this. We did, however, find evidence that somatotroph differentiation is advanced in the PAT animals. We performed immunohistochemistry for GH on e15.5 matched WT (n = 4) and PAT (n = 6) mice. In all PAT individuals we observed a small number of GH^+^ cells, whereas none of the WT sections had GH staining. From this we conclude that somatotroph commitment commences earlier in the *Dlk1*-deficient pituitary. These data were added to Figure 5.

8. Figure 6E – the sox2^+^ MZ volume was calculated using the Cavalieri method. How was the MZ identified in order to distinguish it from the parenchyma?

To be counted as MZ the cell must have at least one membrane edge directly adjacent to the lumen. We recognise that this may slightly underestimate the proportion of the MZ since connectivity to the lumen is not visible in all 2D sections.

9. No description of the methods for RTqPCR is present.

This has now been amended.